



# An 1200-year multi-proxy dendrochronological temperature reconstruction for the area of Austrian Alps

Marzena Kłusek[1,2,3], Michael Grabner[2], Sławomira Pawełczyk[1], Jacek Pawlyta[4]

5    [1]Institute of Physics – Centre for Science and Education, Silesian University of Technology (SUT), ul.
Konarskiego 22B, 44-100 Gliwice, Poland

[2]Institute of Wood Technology and Renewable Materials, University of Natural Resources and Life Sciences,
Vienna (BOKU), Konrad Lorenz-Straße 24, 3430 Tulln an der Donau, Austria

[3]Graduate School "Human Development in Landscapes", Christian-Albrechts-Universität zu Kiel (CAU),
10   Leibnizstraße 3, 24118 Kiel, Germany

[4]Faculty of Geology, Geophysics, and Environmental Protection, University of Science and Technology (AGH),
30 Mickiewicza Avenue, 30-059 Krakow, Poland

*Correspondence to*: Marzena Kłusek (Marzena.Klusek@polsl.pl)

**Abstract.** Temperature reconstruction was carried out on the basis of a centuries-long dendrochronological
scales from Austrian Alps. Chronologies of growth-ring width, maximum density of latewood and stable isotope
content of carbon and oxygen were applied for the research. Subfossil wood and living trees originating from the
area of Schwarzensee Lake were used for the construction of the chronologies. All measurements were
performed with annual resolution. A very good match was found between the results obtained and the
meteorological data, making it possible to precisely reconstruct the temperature of the growing season (May–
September) over the years 800–2000 CE. The proportion of temperature variance explained by independent
variables accounted for 52 % in the period common for the growth-ring chronologies and meteorological data.
The statistics calculated during calibration and verification tests indicated that chronologies have high
reconstruction skills and that the accuracy of reconstruction is good. Obtained data show the existence of
significant cooling in the periods 900–1100 CE, 1275–1325 CE, 1450–1600 CE and 1800–1890 CE and evident
warming around the years 1150 CE, 1250 CE, 1325–1425 CE, 1625–1775 CE. The strongest increasing trend in
temperature has been observed since the beginning of the 20th century and is clearly indicative of an ongoing
climate warming.

## 1 Introduction

Contemporary climate change is having an enormous impact on both human life and the natural environment. In
order to know the scale and amplitude of modern climate fluctuations, it is necessary to trace the history of
weather variations in the past. Such a palaeoclimatic reconstruction also makes it possible to diagnose the causes
of the ongoing warming and to forecast future temperature trends. Consequently, it also provides a basis for
counteracting climate changes or mitigating their impact on the natural ecosystems and the human life.

Wood is an excellent archive of environmental variability because plant tissues record climate conditions
existing during their growth. Weather factors affect the intensity of physiological processes and in this way
influence the growth-ring width, wood density and isotopic signal in wood. The changes of physico-chemical
properties of growth-rings are influenced by solar radiation, temperature, precipitation, and atmospheric
humidity. Fortunately, the dependencies between environmental forces and internal mechanisms taking place in
plants are well understood. As a result, measurements of various wood parameters within individual growth-
rings allow tracking climate variables with an annual resolution and in long-term time intervals.

This article presents a long time scale temperature reconstructions for the Dachstein Mountains area localised in
Northern Limestone Alps in Austria. The reconstruction was carried out on the basis of subfossil wood
excavated from the small mountain lake Schwarzensee and core samples bored from living trees growing around
this lake.





Schwarzensee Lake is situated in mountain area and located almost at an elevation of regional timberline. Therefore, temperature is a dominant control influencing the length of vegetation period and the main factor limiting tree growth in this region. This strong interdependence creates an opportunity for the reconstruction of thermal conditions of the environment. A favourable aspect for palaeoclimatic studies is also the origin of wood from an area characterised by small territorial range. Because all the trees grew on the mountain slopes surrounding the Lake Schwarzensee, they were subjected during the course of their lives to the influences of the same weather conditions (Grabner et al., 2007). This ensures homogeneity of the climate signal recorded in annual growth-rings. Moreover, due to the fact that spruce wood dominated within the material sampled from the Schwarzensee Lake, it was possible to perform analyses on the basis on a single tree species only. This allowed to eliminate potential errors that can result from the various reactions of individual tree taxa to the impact of environmental factors.

In order to make the best usage of wood for dendroclimatic research, various measurements have been performed during previous stages of the work. On this basis, four separate chronologies were built. Particular chronologies were constructed for the ring width (TRW), the maximum density of latewood (MXD) and for the content of stable carbon and oxygen isotopes within individual growth-rings. All these chronologies were exploited during paleoclimate research. The objective of the presented study was to reconstruct the thermal conditions of the environment in the area extending around Lake Schwarzensee, in a period covering the common time range of previously created chronologies. In order to most accurately determine the weather in the past a multi-proxy approach was applied. A combination of several proxies was aimed to increase the amount and the quality of palaeoenvironmental information derived from tree-ring data. Due to the employment of several growth-ring parameters to reconstruct a single environmental variable the reduction of the noise associated with each of the proxies was also obtained and the enhancement of the climatic signal was gained as well (McCarroll et al., 2003).

## 2 Materials and methods

The material for the study came from subfossil wood deposited in the Lake Schwarzensee (47°31′ N, 13°49′ E, 1450 m.a.s.l.) and from trees growing around this lake. Schwarzensee is a small crater-like lake surrounded by steep rocky slopes and cliffs overgrown by forest. Due to the large decline of the slopes, dead trees or trees fallen by wind and snow, moved gravitationally and sank into the water (Kłusek et al., 2015). The sedimentation conditions at the bottom of the lake allowed to remain the logs for hundreds of years in an almost unchanged form. This good preservation state results from the fact that limited availability of oxygen, which is necessary for the development of most decomposition bacteria and fungi, significantly inhibited wood degradation process (Kim and Singh, 2000).

Wood was excavated by dendrochronologists cooperating with a professional team of divers who recruited from the Austrian army. The sampling took place in the summer of the year 1999. During the works, 211 trees were lifted from the bottom of the lake, 66 % of them were spruce (*Picea abies* (L.) Karst.), 21 % larch (*Larix decidua* Mill.) and 13 % belonged to stone pine (*Pinus cembra* L.). Stem discs were cut out from each log and afterwards, sampled trunks were submerged back in the lake's waters. Moreover, 80 spruce and larch samples were bored from the trees growing around the lake in 1999 (Grabner et al., 2007). However, along with the development of subsequent chronologies there was a need for the new samples. Therefore, additional 40 cores from the living trees was collected in years 2009 and 2015.

The wood originating from the lake has been carefully dried. Then, together with the samples coming from the living trees it was prepared for growth-ring measurements. For this purpose, wood transverse section was sanded to obtain clearly visible growth-rings. The ring width was measured with 0.01 mm precision using the LINTAB system and TSAP Win computer program (Rinn, 2003). Obtained ring-width sequences were cross-dated and checked for dating and measurement errors using COFECHA software (Holmes, 1983). As a result of this research a spruce-larch chronology dating back to 1475 BCE was received (Grabner et al., 2007).

The chronologies used during dendroclimatic analysis are much shorter than this original master chronology. This results from the fact that the material intended for densitometric and isotopic studies must meet the relevant requirements. The aquatic conditions at the bottom of the Schwarzensee Lake encouraged proper preservation state of wood, but the decomposition process was not completely inhibited. Along with the time, the degree of wood destruction increased, expressing itself in the physical degradation of wood tissue and in the lower density





of wood (Lamy et al., 2012). Wood decay reflected itself also in the chemical disintegration of organic matter which led, among others, to the changes in the stable isotope content in wood (Sass-Klaassen et al., 2005; Harlow et al., 2006). Therefore, the oldest samples were more destroyed and thus they were omitted during densitometric and isotopic research. In addition, some parts of the samples, such as zones of reaction or juvenile wood that are characterised by disturbed isotopic ratio and disrupted density value had to be eliminated as well. Moreover, only the wood of Norway spruce was selected for dendroclimatic research which also limited the number of samples applied for the research. This decision was taken because various species, even those growing in the same stand, may react in the different way to the influence of weather conditions. All these factors significantly reduced the temporal range of the chronologies in comparison with original master chronology constructed on the basis of Schwarzensee wood.

During the densitometric research, a total of 287 samples (radii) of spruce wood were used, including 50 samples from living trees and 237 from subfossil logs. Thin strips (of thickness 1.2 mm) were cut out from stem discs and from tree cores in such a way that they represented the wood surface perpendicular to the longest axis of the stem – cross-section of the trunk. For the purpose of wood preparation, the samples were rinsed for 24 hours in distilled water at room temperature (Kłusek and Grabner, 2016). Subsequently, the samples were subjected to acclimatisation process which was performed over a 48 h time span at a constant temperature of 20° C and relative air humidity of 65 %. Next, the samples were placed on X-ray sensitive film and exposed to X-raying. For the analysis the Seifert ISO-DEBYEFLEX device was used. Irradiation was performed from a distance of 2.5 m, during a 25 min period, with the accelerating tension of 10 kV and intensity of 24 mA. Afterward, X-ray sensitive films were developed and then digitalised by the means of a dendrodensitometer. Tree-ring transparency (grey level) was obtained at 5 μm intervals along the radial direction of the stem and from these measurements the density of samples was determined by means of computer software (Kłusek et al., 2015).

Chronology of maximum latewood density was constructed with application of the CRUST program (Melvin and Briffa, 2014). Density measurement were standardised using the two-curve RCS method (Melvin et al., 2013). The data for RCS series were split into two equal groups according to higher and lower mean MXD. RCS curves were smoothed with an age dependant spline (Melvin et al., 2007). Moreover, to remove the common climatic signal prior to creating RCS curves the signal free method with maximum 10 iterations was applied for individual measurement sequences (Melvin and Briffa, 2008). Indexes were calculated as ratios and standard chronology was computed as the arithmetic mean. Variance stabilisation of the chronology was performed using Keith Briffa's RBAR weighted method (Osborn et al., 1997). Ring width measurements obtained during densitometric research were compiled into a separate chronology as well. For this purpose the ARSTAN program was utilised (Cook and Krusic, 2005). The parameters selected to chronology development were similar to the previous ones. However, in this case single regional curve (RCS) detrending was adopted. Measurement series were divided by the fitted RCS curve values and chronology was calculated as non-robust, arithmetic mean. The Keith-Briffa RBAR weighted method was chosen to homogenise the variance of chronology (Osborn et al., 1997). From the ARSTAN output, the residual chronology was taken for further analysis. In order to determine the suitability of chronologies for dendroclimatic studies, the commonly used Expressed Population Signal (EPS) was calculated (Wigley et al., 1984; Briffa and Jones, 1990). Signal strength of the TRW and MXD chronologies was assessed using running EPS computed over a moving window of 50 year with 49 year overlap, to reflect the variability of EPS back in time.

For stable isotope analyses 51 subfossil stem discs and samples from five living trees were prepared. Within individual samples growth-rings were split by means of a scalpel. Growth-rings separation was conducted under the magnification of a binocular microscope. Afterwards, each single growth-ring was cut into thin slivers. Then, the material obtained from four trees was pooled (combined) for a particular year. The equal proportion by weight was maintained, and 25 mg were usually weighed out from each growth-ring. For the stable isotope measurements, α-cellulose originated from the whole annual growth-ring was extracted. The procedure of α-cellulose isolation was conducted in accordance with the modified Green (1963) method (Kłusek and Pawełczyk, 2014). During the first phase of chemical treatment water solution of sodium chlorite and 80 % acetic acid was used to remove the lignin from the samples. Next, in order to eliminate the hemicellulose and lignin remains, 10 % and 17 % water solutions of sodium hydroxide were applied. All the steps of extraction process were conducted in ultrasonic bath, usually of 70° C temperature. Only the last reaction with 17 % sodium hydroxide was performed in ultrasonic bath of room temperature. At the final stage of α-cellulose preparation, the samples were ground in a mortar to better separate and homogenise the fibres and then dried at the temperature of 70° C on a hot plate (Kłusek et al., 2019).



Stable isotope ratio was determined for two or three sub-samples of α-cellulose, representing a particular year. The samples were combusted "on-line" at 1020° C during carbon isotope measurements. IAEA cellulose C3 and IAEA Two Creeks wood C5 international standards, and laboratory standard Fluka acid washed cellulose were used as the reference materials, to calibrate the sample isotopic composition versus the international Vienna Pee Dee Belemnite (VPDB). In the case of stable oxygen isotopes the samples were pyrolised "on-line" at temperature 1300° C. IAEA Spruce, Rsta and laboratory standard GdCell were utilised to normalise samples relative to Vienna Standard Mean Ocean Water (VSMOW). Measurements were performed with application of elemental analyser EuroVector directly connected with continuous flow Isotope Ratio Mass Spectrometer IsoPrime EA-CF-IRMS. The precision of this method is 0.1 ‰ and 0.3 ‰ for carbon and oxygen measurements, respectively.

The results obtained during stable isotope measurements were averaged for the particular calendar years and they served directly for the raw chronologies construction. The only exception was the final part of the carbon chronology. For the period of 1820–2000 CE, an additional corrections were introduced. This was aimed to compensate the anthropogenic changes in the concentration of atmospheric carbon dioxide and to counterbalance the human induced impact on the isotopic composition of the air (McCarroll et al., 2009; Konter et al., 2014).

Obtained chronologies of the growth-ring widths, maximum density, stable carbon and oxygen isotopes were compared with climatic records. For this purpose, the temperature data originating from neighbouring meteorological stations and gridded for the area of Schwarzensee Lake were applied (Auer et al., 2005, 2007; Efthymiadis et al., 2006; Chimani et al., 2013). Climate-growth relationships were determined by examining the correlation coefficient (r) between measured tree-ring proxies and temperature. Coefficients were calculated during a 14 month period beginning in September of the previous year and ending in October of the current year, in the time interval 1780–2000 CE. Tests were also performed over various multi-month seasons. Calculations were carried out using the DENDROCLIM 2002 program (Biondi and Waikul, 2004). Next, a linear regression model was developed to calibrate the proxies versus temperature data. The reliability of the transfer function was evaluated by statistics computed for calibration and verification periods. To assess the accuracy of statistical predictions the reduction of error (RE), coefficient of efficiency (CE), Pearson correlation coefficient (r) and coefficient of determination ($R^2$) were determined (Fritts, 1976; Cook et al., 1994; Wahl and Ammann, 2007). All data analyses were done in Reconstats program (Macias-Fauria et al., 2012).

### 3 Results and discussion

On the basis of the conducted research, the chronologies presenting various physicochemical parameters of wood were constructed. The chronology of maximum density and the corresponding chronology of ring width reached back to 88 CE. The stable carbon isotope chronology ranged over 200–2000 CE. In turn, stable oxygen isotope chronology encompassed the period 800–2000 CE. The reconstruction of climatic conditions in the Schwarzensee area was carried out for the time interval common to all chronologies (800-2000; Fig. 1).

To assess the signal strength of the MXD and TRW chronologies the Expressed Population Signal was employed. Usually, for EPS a threshold value of 0.85 is accepted, above which a chronology is assumed to represent the population signal with sufficient confidence for reconstruction purposes (Wigley et al., 1984; Briffa and Jones, 1990). In the case of the ring width chronology, the value of the EPS coefficient was higher than 0.85 during the entire analysed period of 800–2000 CE. However, for the chronology of maximum density in the years 1264 CE, 1365 CE, 1373–1452 CE, 1476–1482 CE, 1576 CE, 1588–1676 CE, 1774–1779 CE, the EPS coefficient was lower than the required value of 0.85. This indicates the reduction in common variance of the chronology during specified years. Therefore, it should be assumed that during these time intervals MXD chronology has a lower predictive power. However, the value of EPS coefficient depends not only on mean inter-series correlation (RBAR) between the samples but it changes also along with the series replication. The observed drop in EPS value for the MXD chronology was connected both with the decrease in the RBAR correlations and was also related to the small number of samples measured for the mentioned time spans. Nevertheless, the EPS coefficient should be treated with caution because it does not reveal whether the population signal is closely related to the reconstructed parameter. This signal could be also linked with any other source of common growth variability (Buras, 2017). Moreover, in the case of RCS chronologies there are also additional factors that could hinder the assessment of signal strength on the basis of the EPS value (Melvin and Briffa, 2014). In contrast to TRW and MXD results, the value of EPS could not be determined for isotopic chronologies because they were built using pooled samples. However, previous studies demonstrated that



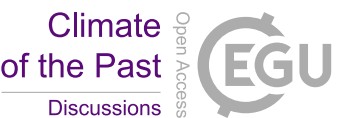

combining growth-rings from four trees before the isotopic measurements, as was the case for both chronologies, provides a representative EPS higher than 0.85 (Leavitt, 2010; Dorado Liñán et al., 2011).

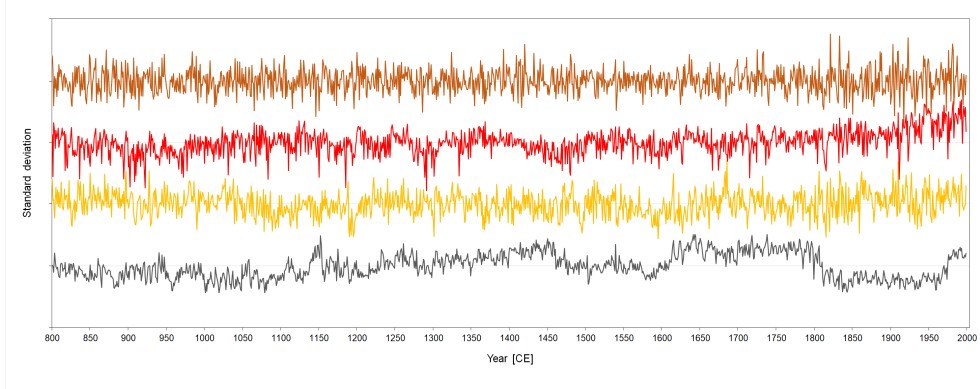

**Figure 1: Normalised chronologies of stable carbon isotope ($^{13}$C–grey line), stable oxygen isotope ($^{18}$O–**
**yellow line), maximum latewood density (MXD–red line), and ring-width (TRW–brown line).**

The obtained chronologies of ring width, maximum density, carbon and oxygen stable isotopes were compared with climatic variables. It was observed that weather parameters influenced individual chronologies in different way and with various strength. However, all measured tree-ring proxies correlated with thermal conditions of the
environment (Table 1). The best results for the ring width chronology were noted for May, June, July and September. In the case of the maximum density chronology, the highest correlations were visible in April–September, and especially during July–September. For both isotope chronologies, the best fit between the temperature and the stable isotope content was obtained for July–August, and in a broader range, for the May–September season. These results prove that all proxies successfully reflect the temperature impact on the growth
of trees.

**Table 1: Bootstrap correlation coefficients calculated in DENDROCLIM 2002 between mean monthly temperature and stable carbon isotope chronology ($^{13}$C), stable oxygen isotope chronology ($^{18}$O), maximum latewood density chronology (MXD), and ring-width chronology (TRW). Significant values at**
**the 0.05 level are marked in bold. Asterisks indicate months of the year preceding growth ring formation. Calculations were performed over the entire period of available meteorological data – 1780–2000 CE. MJJAS – temperature averaged for May–September period.**

| | $^{13}$C | $^{18}$O | MXD | TRW |
|---|---|---|---|---|
| January* | **0.189** | **0.162** | 0.106 | 0.099 |
| February* | 0.116 | 0.090 | 0.078 | 0.046 |
| March* | **0.135** | 0.090 | 0.094 | 0.030 |
| April* | 0.075 | -0.054 | -0.009 | 0.094 |
| May* | **0.202** | 0.033 | -0.015 | -0.042 |
| June* | 0.092 | -0.066 | 0.112 | **-0.227** |
| July* | **0.194** | -0.093 | 0.007 | **-0.185** |
| August* | **0.294** | 0.004 | 0.054 | **-0.186** |
| September* | **0.183** | **0.148** | -0.010 | -0.026 |
| October* | **0.197** | -0.005 | -0.010 | 0.087 |
| November* | 0.036 | -0.015 | 0.028 | 0.033 |
| December* | 0.099 | **0.189** | 0.022 | 0.116 |



| | | | | |
|---|---|---|---|---|
| January | **0.167** | 0.048 | 0.118 | 0.062 |
| February | **0.141** | 0.062 | 0.015 | 0.090 |
| March | 0.104 | 0.118 | **0.161** | 0.008 |
| April | 0.042 | 0.027 | **0.177** | -0.043 |
| May | **0.214** | **0.211** | **0.244** | **0.141** |
| June | **0.135** | **0.187** | 0.054 | **0.399** |
| July | **0.363** | **0.508** | **0.323** | **0.274** |
| August | **0.420** | **0.302** | **0.505** | 0.051 |
| September | **0.158** | **0.151** | **0.359** | 0.135 |
| October | **0.144** | 0.047 | 0.070 | 0.047 |
| November | 0.045 | 0.010 | **0.184** | **0.142** |
| December | 0.066 | -0.018 | **0.134** | 0.012 |
| MJJAS | **0.459** | **0.483** | **0.537** | **0.354** |

There is a lack of clear guidance as to how high the correlation coefficient between meteorological record and
proxies needs to be, to provide reliable results of climate reconstruction. McCarroll and Pawellek (2001)
suggested that correlation should have value at least 0.71 because in this case tree-ring measurements explain
half of the variance in the weather parameter. None of the correlation coefficients calculated between particular
chronologies and the temperature data did not exceed the threshold of 0.71. Therefore, it was decided to combine
proxies in order to improve the retrospective power of the chronologies and in this way to better estimate the past
temperature values. When this method is used, the best results are obtained where the primary control on all
proxies are similar and the other influencing factors are different as in this case they tend to cancel each other.
This is an important issue because if growth-ring parameters are dependent on the same weather variables they
are also subjected to similar sources of error and as a consequence, an application of multi-proxy approach does
not improve the reconstruction skills of chronologies (McCarroll et al., 2003, 2011). The Schwarzensee
chronologies meet the above mentioned requirements and the evidences confirming their suitability for multi-
proxy research are discussed below.

The width of growth-rings depends on various weather parameters. Environmental conditions affect all
physiological processes taking place in the tree. Climate factors control net assimilation and photosynthesis rate,
level of cellular respiration, plant nutrition, hormone functions, transpiration and stomatal conductance as well as
all processes of growth and development on cellular level and at the scale of the whole plant. They significantly
impact division, enlargement and maturation of wooden cells. As a result, temperature, precipitation and solar
radiation could stimulate or inhibit growth-ring increment. For Schwarzensee TRW chronology the strongest
correlation was observed between the width of growth-rings and the temperature values. It was because
temperature is the most important factor limiting the growth of trees in this region (Leal et al., 2007). In contrast
to this, rainfall has a much smaller effect on the width of growth-rings. It is caused by high precipitation level
and an adequate water supply in the area of Schwarzensee.

In the case of densitometric chronology, the main weather parameter causing the changes in maximum latewood
density was temperature as well. Precipitation and sunlight had considerably weaker impact (Kłusek et al.,
2015). Maximum latewood density depends on the cell wall thickness and on cell size in the latewood zone
wherein, the first relationship is much more important (Gindl et al., 2000; Wimmer and Grabner, 2000). In turn,
the thickness of cell walls is controlled mainly by the duration of the cell wall thickening phase (Yasue et al.,
2000). In Alpine region the range of time spent for maturation of latewood cells is mostly conditioned by
temperature of summer months and by the term of cessation of the growing season. Also the length of the
vegetation period is predominantly affected by thermal conditions of the environment. Therefore, the maximum
density of latewood successfully reflects the temperatures of late summer months (Schweingruber et al., 1993;
Wimmer and Grabner, 2000; Gindl et al., 2001).

Similar like for TRW and MXD, for the stable carbon isotope chronology the highest values of correlation were
achieved between the isotopic ratio and the temperature of the vegetation period. Nevertheless, the
correspondence with sunshine level was also strong. It is because the content of stable carbon isotopes in wood
depends on the rate of photosynthesis and on intensity of stomatal conductance. These processes are in turn,
influenced by weather conditions (Treydte et al., 2009; Gagen et al., 2011). During the period of relatively low



temperature, diminished solar radiation and high availability of water, the stomata are widely open and assimilation level is reduced. The carbon dioxide concentration in a leaf is high, because it can diffuse freely into the plant. Moreover, $CO_2$ consumption is limited due to decreased rate of photosynthesis. Then the photosynthetic enzymes use much larger amount of the preferred $^{12}C$. This result in low $^{13}C$ concentration in organic matter formed under these conditions. In contrast to this, when the water resources are insufficient and the air temperature and solar radiation are high, the stomata are almost closed and more intense photosynthesis that takes place in a leaf uses heavier carbon isotope to a larger degree. This happens because carbon dioxide diffusion into a leaf is weaker than the need for photosynthesis and this situation finally causes high $^{13}C$ content in the organic compounds produced during this time (Farquhar et al., 1982; Schleser et al., 1999; Helle and Schleser, 2004). The dominant control for stomatal closure and opening is the vapour pressure deficit (the difference in vapour pressures inside and outside of the leaf). In turn, the assimilation level is mostly related to photosynthetically active radiation (McCarroll and Loader, 2006). Therefore, the temperature signal obtained for Schwarzensee chronology is in the vast majority, indirect. The link observed between the temperature and $\delta^{13}C$ is most probably the consequence of the relationship which exists between thermal conditions and other controlling variables, mainly solar radiation and to a lesser extent vapour pressure deficit (Kłusek et al., 2019).

In comparison with other proxies, the stable oxygen isotope chronology provided a more mixed signal. In the case of $\delta^{18}O$ chronology the highest coefficients of correlation were obtained for temperature but a strong relationship was also observed for the sum of rainfall. In general, the content of stable oxygen isotopes in wood reflects an isotopic composition of water absorbed by the tree and later modified by evapotranspiration in the leaf. The source of water for trees is the soil moisture. The isotopic signature of the soil water depends on evaporation and condensation processes, during which the water enrichment or depletion to heavier oxygen isotopes may arise. Isotope fractionation during the hydrological cycle is strongly associated with air temperature and humidity. On the other hand, xylem water, which moves up the stem and into the leaves, has the same isotope ratio as the source water in the soil. However, before the incorporation of oxygen molecules into the wood tissue, several stages of isotope discrimination take place as well. One of them occurs within the leaf and is caused by the action of the stomata. When the stomata are open the plant loses water and oxygen molecules that comprise the lighter isotopes are preferentially removed. This leads to a relative increase in $^{18}O$ isotope content in the leaf water. Nevertheless, this enlarged concentration is counteracted by the Péclet effect which consists in the convection of isotopically lighter source water through the transpiration flow (Gagen et al., 2011). As a result, the total amount of stable oxygen isotopes in wood depends strongly on the evapotranspiration level which in turn, is reliant on stomatal conductance and leaf to air vapour pressure deficit, both of which are linked to relative humidity of air and to the temperature conditions. In the final effect, strong correlation between temperature and oxygen isotopes is a consequence of a direct impact of thermal conditions on the isotopic composition of water taken up by the plant as well as a result of intermediate influence of temperature on evaporative enrichment in the leaf (McCarroll and Loader, 2004). These processes explain the links observed between stable oxygen isotopes and climate variables in the area of Schwarzensee Lake (Kłusek et al., 2022).

On the basis of presented results, it could be stated that the temperature signal is strong for all chronologies. In turn, the other meteorological factors affecting particular proxies vary in type and strength of influence and therefore, they tend to cancel themselves during application of multi-proxy approach. This is also confirmed by the value of the correlation coefficients calculated between the particular chronologies. Obtained Pearson correlations were poor and the highest value of the correlation coefficient was 0.258 (Table 2). This fact also indicates that the analysed growth-ring parameters reflect the impact of various environmental controls. Nevertheless, all chronologies correlated well with the temperature of individual months during the growing season. These correlations were of different values and the strongest relationships between temperature and particular growth-ring parameters occurred in various time intervals (Table 1). However, for all proxies in the May–September period, correlations were statistically significant. Linear correlation coefficient computed for this season amounted to 0.459, 0.483, 0.537 and 0.354 for carbon, oxygen, MXD and TRW chronologies, respectively. Therefore, for these months, calibration tests were carried out, to determine if the proxies might be combined to provide a more reliable estimate of the common temperature signal. Calculations were performed according to the procedure proposed by McCarroll et al. (2003) within the time span of 1780–2000 CE. For this purpose, each proxy was used independently to estimate May–September temperature on the basis of linear regression model. Obtained results was averaged, with each proxy weighted according to the percentage of its variance that is explained by May–September temperature. The received values were correlated with meteorological data and this allowed to obtain the 'effective correlation' between the estimated and measured values of May–September temperature. The effective correlation accounted for 0.72 and was significantly higher





than the individual coefficients calculated for separate growth-ring parameters. This proves that multi proxy approach provides more accurate results and due to its usage the more reliable reconstruction of past temperature could be obtained for the area of Schwarzensee Lake.


**Table 2: Pearson correlation coefficients calculated between particular chronologies – stable carbon isotope chronology ($^{13}$C), stable oxygen isotope chronology ($^{18}$O), maximum latewood density chronology (MXD), and ring-width chronology (TRW). The time interval 800–2000 CE was used for the comparison.**

|  | $^{13}$C | $^{18}$O | MXD | TRW |
|---|---|---|---|---|
| $^{13}$C |  | 0.050 | 0.033 | 0.055 |
| $^{18}$O | 0.050 |  | 0.258 | 0.241 |
| MXD | 0.033 | 0.258 |  | 0.175 |
| TRW | 0.055 | 0.241 | 0.175 |  |

Nevertheless, for the palaeoclimate reconstruction purposes, it was decided to narrow down the time range of data used. It was possible because meteorological records existing for the Schwarzensee area is extremely long. However, temperature measurements from the eighteenth century and from the first half of the nineteenth century are less precise in comparison with later data (Böhm et al., 2001; Frank et al., 2007) and therefore may adversely affect the quality of palaeoclimate reconstructions. In turn, in the second half of the twentieth century,

the values of growth-ring proxies are disturbed by the influence of human activity (Fonti et al., 2010). Moreover, during this time human induced increase in temperature level is also evident (IPCC, 2022). Both of these reasons cause the changes in the relationship between the growth-ring parameters and climatic factors. This phenomenon was observed directly for Schwarzensee chronologies (Kłusek et al., 2019). As a result, the usage of data burdened with 'anthropogenic noise' would have disadvantageous impact on the reconstruction authenticity.

Therefore, to avoid possible errors, for the validation with meteorological record the period starting since 1840 CE and ending in the year 1959 CE was selected.

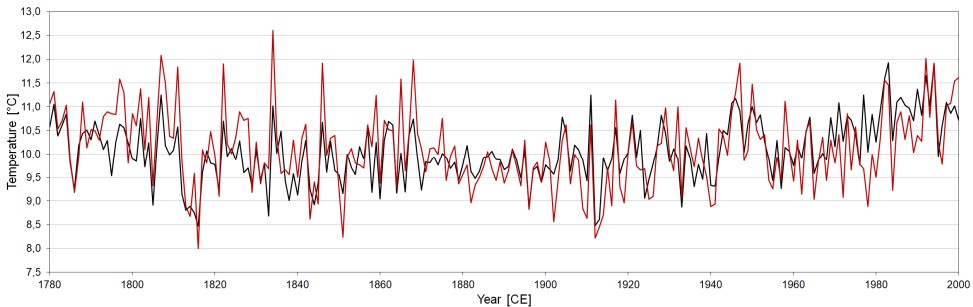

**Figure 2: Comparison of May-September temperature values between meteorological record (red line)**
**and reconstructed data (black line).**

The temporal stability of the climatic signal was determined on the basis of split period validation tests taking advantage of linear regression models and using May–September temperature data. Calibration– verification segments covered the periods 1840–1899 CE and 1900–1959 CE, respectively. To assess the accuracy of the
statistical predictions the reduction of error, coefficient of efficiency, Pearson correlation coefficient and coefficient of determination were calculated (Table 3). Reduction of error and coefficient of efficiency were calculated to evaluate the similarity between the observed and estimated data. The calculated reduction of error was equal to 0.410 (p<0.001), and the coefficient of efficiency amounted to 0.409 (p<0.001). The positive and relatively high value of RE and CE coefficients demonstrates strong reconstruction skills of the chronologies and
is an evidence for a valid regression model. Reduction of error is a measure of shared variance between estimated and observed series and is a highly sensitive indicator of the reconstruction reliability. Reduction of



error ranges from +1 indicating perfect agreement to minus infinity. Commonly, values of RE and CE greater than zero are evidence of a rigorous model. In turn, coefficient of determination was 0.540 (p<0.001) for calibration period and Pearson correlation coefficient had value of 0.546 (p<0.001) for verification period. These
scores prove that a linear association between measured and estimated variables is robust and that the regression predictions well approximate the real meteorological data (Fig. 2).

**Table 3: Statistics calculated in Reconstats program used to evaluate the quality of the model applied for climate reconstruction – reduction of error (RE), coefficient of efficiency (CE), coefficient of**
**determination ($R^2$), and Pearson correlation coefficient (r) explained variance adjusted for loss of degrees of freedom ($r^2$). Full validation period: 1840–1959, calibration period: 1840–1899, verification period: 1900–1959. Third column presents significances of the statistics in the 95 % and 99 % confidence intervals.**

| Validation period: | RE=0.410 (p<0.001) | RE_95=0.046, RE_99=0.084 |
|---|---|---|
| Validation period: | CE=0.409 (p<0.001) | CE_95=0.023, CE_99=0.056 |
| Calibration period: | $R^2$=0.540 (p<0.001) | $R^2$_95=0.161, $R^2$_99=0.207 |
| Verification period: | $r^2$=0.546 (p<0.001) | $r^2$_95=0.065, $r^2$_99=0.111 |

These statistics obtained during validation tests point that reconstruction skills of chronologies are high. Therefore, for the entire period of 1840–1959 CE a linear regression model was calculated to reconstruct May–September temperature in the time range covered by all Schwarzensee chronologies, thus for the years 800–2000 CE. Coefficient of determination received during calculation of regression model shows that the proportion of temperature variance explained by independent variables accounted for 52 % in the period common for the
growth-ring chronologies and meteorological data – $R^2$=0.518, significance level p<0.001, calculated for full validation period.

The obtained temperature reconstruction (Fig. 3) indicates the existence of distinct cooling and warming periods in the area of Schwarzensee Lake. The episodes characterised by an increase in temperature level occurred around the years 1150 CE, 1250 CE, and in the periods 1325–1425 CE, 1625–1775 CE. The global warming at
the end of the twentieth century is also clearly evident in the Schwarzensee region. It started in 1925 CE and intensified since 1975 CE. In turn, the temperature reduction in this area prevailed during the time intervals 900–1100 CE, 1275–1325 CE, 1450–1600 CE and 1800–1890 CE.

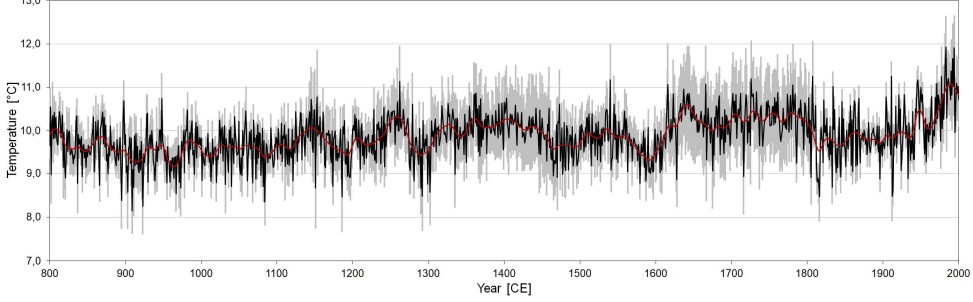

**Figure 3: Schwarzensee May-September temperature reconstruction (black line) and the reconstructed values smoothed using spline fit with 50% variance cutoff at a wavelength of 25 years (red line). Grey shade represents lower and upper 95% confidence interval values calculated in Reconstats program.**

These results reflects the temperature conditions at a relatively small territory localised close to Lake
Schwarzensee. This is due to the fact that all of the material used to the construction of individual chronologies originated from a very small area surrounding the lake. This is a beneficial factor for conducting



palaeoenvironmental research because it ensures homogeneity of the climate signal. At the same time, however, it also causes that reconstruction has a local scale. This is clearly visible especially in relation to the Alps, because in this region there is a large spatial variability of weather conditions (Böhm et al., 2001). Therefore, the

temperature value and the sum of precipitation can vary strongly within the sites situated close to each other. In order to identify whether obtained reconstruction represents climate trends that are coherent over a larger spatial scale, the comparison between obtained results and other tree-ring based temperature reconstructions from nearby Alpine regions was carried out (Fig. 4).

For this purpose, the temperature reconstruction from the whole Alpine region was used (Trachsel et al., 2012).
This reconstruction covered the temperature of June, July and August in the period 1053–1996 CE and was carried out using tree-ring width, maximum density and lake sediment data. According to this research summer temperatures of the last millennium were characterised by two warm and two cold phases. The warming time intervals were defined as 1053–1171 CE and 1823–1996 CE, and cooling tendencies were assigned to 1172– 1379 CE and 1573–1822 CE. In comparison with the results presented in this article, both similarities and

discrepancies are evident. In both reconstructions, there is an increase in temperature around the year 1150 CE, but in the case of reconstruction carried out for the whole area of Alps, in this time the maximum value of temperature occurs, that correspond to the strongest warming in the entire millennium, except only the temperature enlargement in the 20th century (Trachsel et al., 2012). However, for the Schwarzensee region in the later centuries, the temperature has a higher level. Temperature rising appears in both reconstructions also

around the years 1400 CE and 1650 CE. A warming trend between 1600 CE and 1800 CE is visible in both of them as well. A rapid heating which begins around the year 1925 CE is characteristic for Schwarzensee and for the whole Alps.

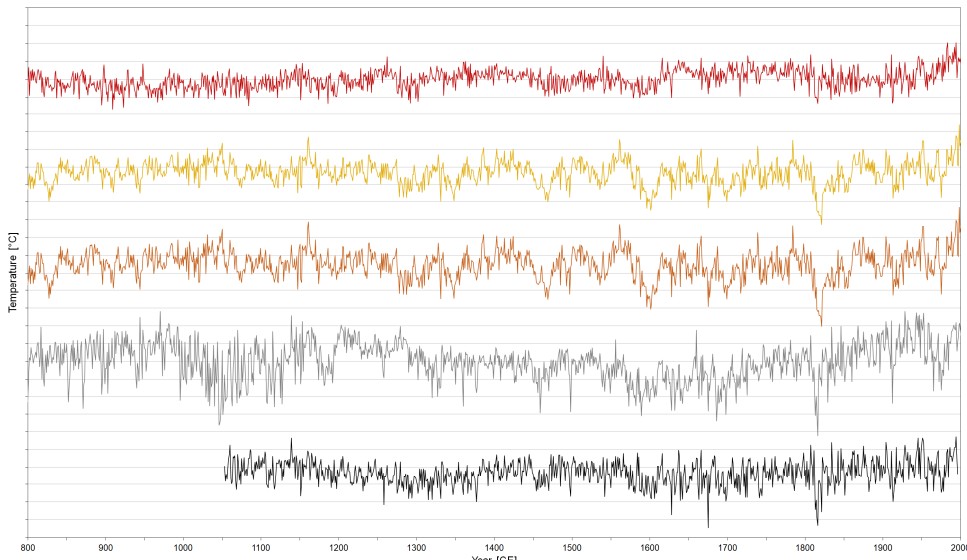

**Figure 4: Comparison of the temperature reconstruction from Schwarzensee area with the other research. Red line–May–September temperature reconstruction based on wood from the Schwarzensee lake area; yellow line–temperature anomalies for June–August for European Alps reconstructed on the basis of tree- ring width (Büntgen et al., 2016); brown line–temperature anomalies for June–August for Austrian Alps reconstructed on the basis of tree-ring width (Büntgen et al., 2011); grey line–temperature anomalies for**
**June–September for Swiss Alps reconstructed on the basis of tree-ring maximum density (Büntgen et al., 2006); black line–temperature anomalies for June–August for whole Alps reconstructed on the basis of tree-ring width, tree-ring maximum density and lake sediments (Trachsel et al., 2012).**





In the case of the reconstruction carried out for the whole Alpine region, the lowest temperatures were recorded in the 14th century, at the end of the 16th century and during the 17th century (Trachsel et al., 2012). For the Schwarzensee area in the 14th and 17th centuries there are opposite trends and occurs an increase in temperature level. On the other hand, the temperature reduction at the end of the 16th century is also clear in the Schwarzensee area. Similar temperature drops occur in both reconstructions also around 1300 CE and 1825 CE. Moving correlation coefficient calculated within 50-year window and using 49 year overlap shows that the
highest resemblance between these reconstructions exists in the time span of the years 1100–1200 CE, 1240–1270 CE, 1450–1600 CE and from 1640 CE to the end of twentieth century. During these time periods correlation coefficient is above 0.5. In turn, the smallest convergence between both reconstructions occurs before 1100 CE and during the years 1280–1420 CE. Nevertheless, average correlation coefficient calculated for the whole range of both time series amounts to 0.474. This indicates a good match of the analysed reconstructions.

The results obtained for Schwarzensee area are also similar to the temperature reconstruction performed for June–August, and based on the chronology of the annual growth-ring width (Büntgen et al., 2011, 2016). The wood used for the construction of this chronology came from high-altitude sites located in the area of the Austrian Alps and in a close neighbourhood of Schwarzensee Lake. The similarity between these data expresses itself especially in low frequency domain. The temperature decrease was observed in both reconstructions
around the years 900 CE, 1300 CE, 1450 CE, 1600 CE, 1800 CE. In contrast to this, the temperature increase occurred in both cases around the years 1150 CE, 1400 CE, 1550 CE, 1650 CE, and 1925 CE.

A somewhat smaller conformity exists between the results received for the Schwarzensee area and the reconstruction obtained for the Swiss Alps (Büntgen et al., 2006). The Swiss chronology was built with application of measurements of latewood maximum density. This reconstruction covers the months from June to
September within the time range of 755–2004 CE and shows the temperature drops around the years 1050 CE, 1110 CE, 1190 CE, 1325 CE, 1450 CE, 1600 CE, 1650 CE, 1690 CE, 1810 CE, 1975 CE, while increases are visible around the years 975 CE, 1150 CE, 1200 CE, 1275 CE, 1475 CE, 1800 CE, 1950 CE. These data prove that Swiss Alps reconstruction only partially coincide with the reconstruction obtained for Schwarzensee area.

Taking into account the reconstruction performed in large spatial scale which covers the entire Northern Hemisphere (Esper et al., 2018), there are less convergences and more differences in comparison with the results obtained for the area of Schwarzensee Lake. It is clearly noticeable that with the increasing distance between the reconstructions, the similarity between them decreases. Discrepancies in reconstructed temperature values also result from various time intervals analysed during particular research. In the case of the Schwarzensee area, the temperature reconstruction concerns the months from May to September. In turn, in reconstructions obtained for
the entire Alpine area (Trachsel et al., 2012) and for the Austrian Alps (Büntgen et al., 2011, 2016) the months from June to August were utilised. Whereas, for reconstruction originating from the Swiss Alps (Büntgen et al., 2006) the period June–September was analysed.

Not without significance is also the usage of various tree species and different wood parameters such as growth-ring width, maximum latewood density, content of stable isotopes, or incorporation of lake sediment data, as it
was in the case of reconstruction presented by Trachsel et al. (2012). While only spruce wood was used to construction of Schwarzensee chronologies, other reconstructions were based on larch wood (Büntgen et al., 2006) pine and larch samples (Büntgen et al., 2011, 2016) and wood of larch, pine, spruce and other coniferous genera (Trachsel et al., 2012). This can be to some extent a source of disagreements between compared reconstructions because each tree species reacts to the influence of weather factors and adapts to changes
occurring in the environment in a specific way. Nevertheless, despite observed differences, there are also clear similarities between compared temperature reconstructions and this testifies to the correctness of obtained results.

## 4 Conclusions

The article presents reconstructions of temperature conditions in the Alpine region localised in Dachstein Mountains. Subfossil and modern wood samples originated from the area of Schwarzensee Lake was used for this study. For the purpose of the palaeoclimate studies, the measurements of growth-ring width, maximum latewood density and stable carbon and oxygen isotopes content, were carried out. On the basis of these measurements four separate chronologies, characterised by annual resolution, were built. During the construction
of these chronologies special attention was paid to retain long-term variance expressed in multi-decadal and





multi-centennial scales. Therefore, RCS standardisation curves were applied for the construction of TRW and MXD chronologies. The usage of isotopic data also contributed to this goal because isotopic sequences maintain climatic signal in the low frequency domain.

Individual chronologies obtained during performed research covered various time spans. However, the common period for all chronologies was years 800–2000 CE. Therefore, for this time interval a temperature reconstruction was carried out. Each chronology reflected the impact of different atmospheric factors on tree growth. Nevertheless, all of them were dependent on thermal conditions of the environment. The temperature signal was strong for all chronologies, while the other meteorological parameters affecting particular proxies differed from one another. Owing to this, the usage of several proxies allowed to improve the precision of 485 conducted research. Due to the combination of few physicochemical parameters of wood, more detailed temperature reconstruction was possible. The statistics calculated during calibration and verification tests indicated that the chronologies have high reconstruction skills and that the accuracy of reconstruction is good. In order to additionally verify the correctness of obtained results and to check their similarity in relation to other research, a comparison was made with other dendroclimatic data published so far. This overview indicates the 490 need to create a multiple long-term local and regional chronologies which could be used for palaeoclimatic reconstructions at a larger spatial scale.

**Author contribution**

MK and MG conceived the ideas and designed the methodology. MK, MG, SP and JP measured the samples and 495 analysed the data. MK prepared the manuscript with contributions from all co-authors. All authors contributed critically to the drafts and gave final approval for publication.

**Competing interests**

The authors declare that they have no conflict of interest.

**Acknowledgements**

This work was funded by the Austrian Science Fund FWF [grant numbers M 1127-B16, P 23998-B16] and based on the collaborative work of the CRC 1266, supported by the Deutsche Forschungsgemeinschaft (DFG, German Research Foundation)—Project-ID 290391021—SFB 1266 and under Germany's Excellence Strategy—EXC 2150—390870439.

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
