# Peer review of "An 1200-year multi-proxy dendrochronological temperature reconstruction for the area of Austrian Alps"

_Climate of the Past, 2024_

## Author Comment (AC1)

Reply to the Reviewer 1:

Thank you very much for revising the manuscript and for valuable comments, which significantly improved the article. In the course of correcting the manuscript, we strictly followed the suggestions provided by the Reviewer.

Revisions have been made in response to particular comments:

***General comments****:*

*Figures should be enhanced by including units on the y-axis.*

Figures 2 and 6 have been enhanced with units on the y-axis (see below).

*Several statements need to be supported by citations to existing literature or the current results. The introduction requires more background detail and citations of previous dendroclimatological studies in the region.*

More background details and citations from previous dendroclimatological studies have been added to the Introduction chapter:

This article presents a long time scale temperature reconstructions for the Dachstein Mountains area localised in Northern Limestone Alps in Austria. The Alpine region has been used for palaeoclimatic studies since many decades. This is the result of beneficial factors that favour the undertaking of such research. The most important of these is the influence of the thermal conditions of the environment, which significantly limit the growth of plants and thus enable temperature changes to be reflected in the annual tree-rings. From the Alpine area predominantly originate dendroclimatic reconstructions based on tree-ring width and the maximum latewood density. Among them are also multi-century analyses, extending back many hundreds of years (Schweingruber et al., 1987, 1988; Büntgen et al., 2005, 2006; Esper et al., 2007; Corona et al., 2010, 2011; Kuhl et al., 2024). On the other hand, chronologies based on stable isotope measurements have been constructed less frequently to date (Treydte et al., 2001; Gagen et al. 2006; Hafner et al., 2014). Moreover, they rarely reach back a thousand years (Kress et al., 2014). However, more than a millennium-old stable isotope studies are known from other territories (Churakova-Sidorova et al., 2022; Büntgen et al., 2021). In turn, in Alpine area reconstructions that use dendrochronological data together with other proxies, such as lake sediments, also occur (Trachsel et al., 2012). Sometimes historical information is used to trace climate fluctuations in the past as well (Casty et al., 2005; Meier et al., 2007).

*Currently, only correlations with temperature are presented for each proxy. The analysis should be expanded to include an examination of the relationships with other climate variables, such as precipitation and drought indices.*

In addition to correlations with temperature, the analysis has been extended to examine relationships with other climatic variables:

Obtained chronologies of the tree-ring width, maximum latewood density, stable carbon and oxygen isotopes were compared with climatic records. For this purpose, various meteorological parameters were applied. The mean monthly temperature (TEM) data originated from neighbouring meteorological stations and was gridded for the area of Schwarzensee Lake (Auer et al., 2005, 2007; Efthymiadis et al., 2006; Chimani et al., 2013). Homogenised mean monthly sunshine duration (RAD) derived from the meteorological station of Kremsmünster (Auer et al., 2007). Total monthly precipitation (PRE) dataset was based on homogenized precipitation series from adjacent meteorological stations and was gridded at 10-minute resolution (Efthymiadis et al., 2006). Self-calibrating Palmer Drought Severity Index (scPDSI) was calculated using as input the interpolated monthly

precipitation and temperature observations with a resolution of 10 minutes longitude by 10 minute latitude (van der Schrier et al., 2007). Vapour pressure (VAP) record contained the spatial monthly averages calculated using area-weighted means for Austria. It was based on the CRU TS high-resolution multivariate gridded data set (Harris at al., 2020). The Standardized Precipitation-Evapotranspiration Index (SPEI) came from the Global SPEI database which offers long-time, robust information about drought conditions with a 0.5 degrees spatial resolution and a monthly time resolution. It is based on monthly precipitation and potential evapotranspiration data from the Climatic Research Unit of the University of East Anglia. This database provides SPEI time-scales between 1 and 48 months and SPEI time-scale used during presented study was 1 month (Beguería et al., 2014). In total, the meteorological records employed for this research encompassed monthly averaged data for temperature (1780–2000), solar radiation (1884–2000), monthly sums of precipitation (1800–2000), monthly averaged self-calibrating Palmer Drought Severity Index (1800–2000), vapour pressure (1901–2000), and Standardized Precipitation-Evapotranspiration Index (1901–2000).

*While the comparison with other temperature reconstructions is interesting, the unique contribution of this multi-proxy chronology compared to reconstructions based on a single proxy needs to be clarified. Namely, what specifically does the combination of TRW, MXD and stable isotopes provide beyond TRW or MXD alone?*

**"The difference may be in the low frequency or long-term trend."**

Information has been added explaining the benefits resulting from using different tree-ring parameters for reconstruction - stable isotopes of carbon and oxygen beyond TRW and MXD alone:

An undeniable advantage of the analyses presented here is the employment of a number of various proxies, including carbon and oxygen stable isotope chronologies, which allow low-term weather trends to be preserved. In comparison with them, the most commonly used tree-ring width or maximum latewood density reflect more temperature fluctuations in the high-frequency domain. Therefore, the applied approach makes it possible to trace thermal changes more accurately over millennial time span and, consequently, to better place contemporary climate warming in historical framework. Results demonstrated here has therefore the potential to improve temperature reconstruction and its temporal reliability. This constitutes an exceptionally favourable situation that distinguishes this work from most previous studies conducted using chronologies of tree-ring width and maximum density of latewood.

*The climate analysis and comparison with other records should be improved. References to major climate periods or events (e.g. Late Antique Little Ice Age, Medieval Climate Anomaly, Tambora Eruption)* **could put the temperature fluctuations in a broader context**.

References to major climate periods and volcanic events were added:

In the obtained reconstruction the effect of volcanic events which could significantly reduce temperature value is also evident. Despite the fact that the area of Schwarzensee Lake is located far from volcanically active regions, however, severe eruptions periodically alter weather conditions, even on a global scale. Taking into account only strong events characterized by an explosivity index equal to or greater than 4 (Siebert and Simkin, 2002) and the fifty coldest years in the entire Schwarzensee temperature reconstruction, 13 of these are could be related to the impact of volcanoes. Although it is rather difficult to unambiguously ascribe cold years to a particular eruption however, probably temperature decrease which appeared in years 1020, 1032, 1040, 1281, 1290, 1587, 1813, 1814, 1815, 1816, 1912, 1913, 1933 were triggered by them. Observed interrelationship is apparent especially for the younger part of the data because for earlier time periods there is a lot of uncertainty due to imprecise dating of the oldest volcanic eruptions. The above comparison is based on the data available on the website https://volcano.si.edu/.

Other global climate fluctuation, such as the Medieval Warm Period or the Little Ice Age, can also be identified in the reconstruction from Lake Schwarzensee. The time span of temperature increase that coincides with the

Medieval Warm Period is evident and covers the years 975–1275 CE. In contrast, the period of the Little Ice Age, defined approximately between 1450 and 1850 CE, is shorter in the Schwarzensee temperature reconstruction and comprises the years 1475–1625 CE. However, on the basis of the research carried out, it is difficult to determine the reason. This shift might be induced by the stable carbon isotope chronology, as it shows an increasing trend between 1600 and 1800 CE, maybe related to the fact that the stable isotope carbon chronology, in addition to temperature, depends strongly on the solar radiation of the summer months. The cause of the upward tendency in reconstructed temperature between 1650 and 1800 CE may also be moisture conditions as the time interval between 17th and 19th centuries was a wet period and all the chronologies used for reconstruction correlate to a greater or lesser extent with such climate parameters as precipitation, VAP, scPDSI, and SPEI indexes (Table 1).

The drivers may be also the range of months employed in the temperature reconstruction (May–August), which differs from many others previously performed studies, often taking into account only the summer period. The lack of perfect matching between the commonly used dates of Little Ice Age and the reconstruction derived from Lake Schwarzensee may also be the result of local environmental conditions that did not coincide with global temperature changes. The Intergovernmental Panel on Climate Change Third Assessment Report concluded that the Little Ice Age is largely independent regional climate change, rather than a globally synchronous increased glaciation and the timing and the areas affected by temperature decrease vary significantly (IPCC, 2022). The differences may also be explained by the combination of multiple proxies in the Schwarzensee reconstruction, in particular the usage of stable isotopes of carbon and oxygen, whose chronologies have the potential to capture more low-frequency variability compared to chronologies based on tree-ring width and maximum latewood density, which in turn, focus on the variability in the short-term domain. It should be emphasised that the undeniable advantage and innovative aspect of the presented research is the application of four different chronologies owing to which Schwarzensee reconstruction has the unique potential to reflect long-term temperature trends.

*Stable isotopes have the potential to capture lower frequency variability compared to TRW and MXD, but this potential is not fully explored.*

In the revised version of the manuscript, the potential of the Schwarzensee reconstruction to capture long-term temperature trends is emphasised several times, also when discussing the differences between the presented results and other palaeoclimatic reconstructions, as well as global trends in temperature change such as the Medieval Warm Period and Little Ice Age.

*Comparisons could be made with additional non-tree ring reconstructions, such as Casty et al. 2005 (https://doi.org/10.1002/joc.1216) and Meier et al. 2007 (https://doi.org/10.1029/2007GL031381).*

The suggested studies, i.e. Casty et al. 2005 and Meier et al. 2007, were mentioned in the Introduction chapter, but a more detailed comparison of temperature reconstructions was made only with reference to the most similar dendrochronological results. A comparison with many previous palaeoclimatic studies would have increased the volume of the article too much, disrupting its proportions. In this article, the authors wanted to present mainly the research results obtained, and broader comparisons as well as reconstructions based on the various research results could be the subject of further papers.

**Specific comments**:

*Lines 38-39: Note that soil moisture is also an important driver of tree growth.*

Required information has been added:

The changes of physio-chemical properties of growth-rings are influenced by solar radiation, temperature, precipitation, soil moisture, and atmospheric humidity.

*Lines 43-55: Consider adding a figure with a map showing the geographical location to provide more context.*

Map showing the geographical location of Schwarzensee Lake has been added:

[Figure]

**Figure 1: Map showing the geographical location of Schwarzensee Lake. Google Maps (2024)**
**https://www.google.pl/maps/place/47%C2%B031'00.0%22N+13%C2%B049'00.0%22E/@47.5166703,12.4983074,8z/data=!4m4!3m3!8m2!3d47.5166667!4d13.8166667?hl=pl&entry=ttu**

*Line 46: Clarify the elevation site.*

Required information has been added:

Schwarzensee Lake is situated in mountain area and located almost at an elevation of regional timberline (47°31′ N, 13°49′ E, 1450 m.a.s.l.).

*Lines 85-86: Explain why Figures 2, 3, and 4 end in the year 2000 when the study goes to 2018.*

Required explanation has been added:

Moreover, 80 spruce and larch samples were bored from the trees growing around the lake in 1999 (Grabner et al., 2007). However, along with the development of subsequent chronologies there was a need for the new samples. Therefore, additional 40 cores from the living trees was collected in years 2009 and 2015. Nevertheless

in this case, the increment cores taken had an smaller inner diameter than during first sampling carried out using dry wood borers. Hence, the climatic reconstruction presented here reaches to the year 2000 CE, as some of the measurements were conducted using cores of bigger diameter only due to the larger amount of wood available for preparation.

*Lines 135-137: It would be helpful to show the EPS, r-bar statistics in a supplementary figure.*

Figures showing EPS and r-bar statistics have been added:

[Figure]

**Supplementary Figure 1: Expressed Population Signal (EPS) and mean inter-series correlation (Rbar) values calculated for the Schwarzensee maximum latewood density (MXD) data set employed for the chronology created using two-curve, signal-free RCS method showing EPS (black) and Rbar (red) computed over 50-year windows, lagged by 25 years. Chronology of maximum latewood density was constructed with application of the ARSTAN program (Cook and Krusic, 2005).**

[Figure]

**Supplementary Figure 2: Expressed Population Signal (EPS) and mean inter-series correlation (Rbar) values calculated for the Schwarzensee tree-ring width (TRW) data set employed for the chronology created using one-curve, RCS method showing EPS (black) and Rbar (red) computed over 50-year windows, lagged by 25 years. Chronology of tree-ring width was constructed with application of the ARSTAN program (Cook and Krusic, 2005).**

*Lines 215-216: The finding of higher correlations between TRW and temperature in June and September agrees with previous work such as Leal et al. 2007.*

The sentence has been added:

The finding of high correlations between TRW and temperature in June and July agrees with previous work such as Leal et al. 2007.

*Lines 230-240: The moderate correlation coefficients may be due to the long-time windows used. Consider analysing 1900-2000 to demonstrate stronger correlations, using the remaining period for verification. Correlations further back in time can be less precise also for a decrease of the observation quality.*

Climate analysis for the period 1901-2000 CE has been added and is shown in Table 1:

| | δ¹³C | | | | | | δ¹⁸O | | | | | | MXD | | | | | | TRW | | | | | |
|---|---|---|---|---|---|---|---|---|---|---|---|---|---|---|---|---|---|---|---|---|---|---|---|---|
| | TEM | PRE | VAP | PDSI | SPEI | RAD | TEM | PRE | VAP | PDSI | SPEI | RAD | TEM | PRE | VAP | PDSI | SPEI | RAD | TEM | PRE | VAP | PDSI | SPEI | RAD |
| January | -0,22 | 0,01 | -0,18 | 0,11 | -0,05 | 0,18 | 0.05 | 0.10 | 0.08 | 0.15 | 0.07 | 0.00 | 0.12 | -0.10 | 0.09 | 0.07 | -0.10 | 0.02 | -0.06 | 0.18 | -0.07 | 0.14 | 0.17 | -0.14 |
| February | 0,01 | **0,22** | 0,04 | 0,20 | 0,20 | -0,11 | -0.01 | 0.08 | 0.03 | 0.10 | 0.00 | 0.06 | -0.02 | 0.07 | -0.02 | 0.03 | -0.01 | 0.04 | 0.12 | 0.09 | 0.13 | 0.19 | 0.11 | -0.08 |
| March | -0,12 | 0,09 | -0,07 | 0,11 | 0,13 | -0,05 | 0.02 | -0.09 | 0.02 | 0.01 | 0.01 | -0.03 | 0.13 | 0.09 | 0.04 | -0.02 | 0.07 | 0.04 | -0.09 | 0.07 | 0.01 | 0.11 | 0.06 | -0.16 |
| April | 0,03 | 0,10 | 0,08 | 0,18 | 0,03 | 0,19 | 0.17 | -0.04 | 0.15 | -0.04 | -0.06 | 0.09 | **0.26** | -0.11 | **0.22** | -0.08 | -0.17 | **0.27** | 0.00 | 0.08 | 0.07 | 0.15 | 0.07 | -0.04 |
| May | 0,14 | -0,20 | -0,04 | 0,09 | -0,25 | 0,17 | **0.30** | -0.11 | **0.24** | -0.09 | **-0.21** | **0.23** | **0.35** | **-0.21** | **0.22** | -0.14 | **-0.31** | 0.17 | **0.16** | -0.14 | 0.10 | 0.06 | -0.12 | 0.17 |
| June | 0,19 | -0,12 | 0,04 | 0,00 | -0,15 | **0,28** | 0.10 | -0.04 | -0.07 | -0.16 | -0.06 | 0.11 | 0.13 | 0.09 | 0.17 | -0.09 | 0.10 | -0.15 | **0.29** | -0.10 | 0.17 | -0.05 | -0.16 | **0.35** |
| July | **0,42** | -0,15 | **0,35** | -0,08 | **-0,26** | **0,54** | **0.50** | **-0.52** | 0.19 | **-0.35** | **-0.54** | **0.65** | **0.45** | **-0.20** | **0.30** | -0.16 | **-0.32** | **0.39** | **0.23** | -0.08 | 0.12 | -0.05 | -0.11 | **0.27** |
| August | **0,51** | **-0,30** | 0,25 | -0,18 | **-0,27** | **0,41** | **0.35** | -0.19 | 0.10 | **-0.37** | **-0.19** | **0.25** | **0.67** | -0.13 | **0.51** | **-0.22** | **-0.24** | **0.27** | 0.02 | -0.12 | -0.07 | -0.11 | -0.08 | 0.06 |
| September | -0,05 | 0,01 | -0,08 | -0,10 | -0,11 | -0,03 | 0.06 | -0.10 | 0.09 | **-0.33** | -0.07 | 0.05 | **0.48** | **-0.37** | **0.36** | **-0.30** | **-0.38** | **0.31** | 0.09 | -0.09 | 0.05 | -0.11 | -0.13 | 0.08 |
| October | 0,03 | 0,09 | 0,00 | -0,03 | 0,07 | -0,01 | 0.01 | -0.06 | -0.03 | **-0.31** | -0.04 | 0.15 | -0.02 | 0.08 | -0.08 | -0.19 | 0.03 | **0.19** | 0.11 | -0.07 | -0.01 | -0.16 | -0.11 | 0.10 |
| November | -0,23 | 0,20 | -0,09 | 0,11 | 0,22 | -0,17 | -0.11 | 0.13 | -0.16 | -0.18 | 0.08 | -0.05 | 0.19 | 0.10 | 0.09 | -0.11 | 0.13 | **0.26** | 0.04 | 0.07 | 0.00 | -0.13 | 0.08 | 0.06 |
| December | 0,11 | -0,08 | 0,16 | 0,07 | -0,04 | -0,07 | -0.03 | 0.10 | 0.07 | -0.10 | 0.08 | -0.13 | 0.10 | 0.11 | 0.03 | 0.02 | 0.13 | -0.02 | -0.03 | -0.18 | -0.04 | -0.14 | -0.12 | -0.03 |

**Table 1. Correlations with climate parameters. Bootstrap correlation were calculated in DENDROCLIM 2002 between climate data: mean monthly temperature (TEM), total monthly precipitation (PRE), mean monthly sunshine duration (RAD), monthly averages of vapour pressure (VAP), self-calibrating Palmer Drought Severity Index (PDSI), and Standardized Precipitation-Evapotranspiration Index (SPEI), and tree-ring parameters: Schwarzensee tree-ring width (TRE), maximum latewood density (MXD), stable carbon ($\delta^{13}$C), and oxygen ($\delta^{18}$O) isotope chronologies. Significant values at the 0.05 levels are marked in bold. Analyses were undertaken over a common period 1901–2000 CE.**

*Also, I suggest examining correlations with precipitation, humidity, scPDSI, and SPEI using interpolated CRU data for 1900-2000.*

Required information has been added:

The obtained chronologies of tree-ring width, maximum latewood density, carbon and oxygen stable isotopes were compared with climatic variables. For this purpose, correlation coefficients were calculated between the individual chronologies and temperature, precipitation, solar radiation, VAP, SPEI, scPDSI variables. In order to be able to better compare the results obtained, all calculations were carried out over the period 1901-2000 CE, as this was the maximum range of the chronologies used for the study and part of the climate data. The results are presented in Table 1. As an effect of this calculations, it was observed that weather parameters influenced individual chronologies in different way and with various strength. Because temperature fluctuations over time are the most interesting parameter in the context of ongoing climate change, a reconstruction of the thermal conditions of the environment was therefore carried out in the work presented here. Temperature is also the only parameter that significantly affects all analysed proxies, so all chronologies from the Schwarzensee Lake area could be used for its reconstruction. This decision made it also possible to apply a multi-proxy approach in the study. A combination of several proxies was aimed to increase the amount and the quality of palaeoenvironmental information derived from tree-ring data. Due to the employment of several tree-ring parameters to reconstruct a single environmental variable it was possible to reduce the noise associated with each of the proxies and to enhance the climate signal. Moreover, the incorporation of stable isotope chronologies into research allowed better identification of reconstructed long-term trends, compared to studies using only TRW or MXD, which better reflect the changes in the high frequency domain.

*Lines 234-235: Show the climate correlation analysis between temperature and the final combined proxy chronology.*

Required Figure has been added:

[Figure]

**Figure 3: Moving interval bootstrap correlation coefficients computed in DENDROCLIM 2002 program for mean monthly temperature averaged for May–September period calculated between meteorological record and reconstructed data. A base length of 25 years was progressively slid through the years from 1800 CE to 2000 CE. The figure shows significant values at the 0.05 level.**

*Line 247: Show precipitation and hydroclimate correlation analyses to support this statement.*

Required correlation analysis has been added in Table 1:

For Schwarzensee TRW chronology the strongest correlation was observed between the width of tree-rings and the temperature and solar radiation values (Table1).

*Line 265: Consider showing this correlation analysis in a supplementary figure.*

Required correlation analysis has been added in Table 1:

Similar like for TRW and MXD, for the stable carbon isotope chronology the high values of correlation were achieved between the isotopic ratio and the temperature of the vegetation period. Nevertheless, the correspondence with sunshine level was also strong (Table 1).

*Lines 263-280: Good explanation of temperature signal, however previous Alpine isotopic studies found stronger drought signals. Show the correlations with all variables to prove temperature signal.*

The correlations with various meteorological variables is shown in Table 1.

*Line 283: Show the summer precipitation correlation analyses.*

Required correlation analysis has been added in Table 1:

In comparison with other proxies, the stable oxygen isotope chronology provided a more mixed signal. In the case of $\delta^{18}O$ chronology the high coefficients of correlation were obtained for temperature but a strong relationship was also observed for total monthly sum of precipitation, monthly averages of vapour pressure, self-calibrating Palmer Drought Severity Index, Standardized Precipitation-Evapotranspiration Index and mean monthly sunshine duration (Table 1).

*Line 314: State "significant correlations" to emphasize if the correlations are significant or not.*

The sentence has been corrected:

Linear correlation coefficient computed for this season was significant and amounted to 0.459, 0.483, 0.537 and 0.354 for carbon, oxygen, MXD and TRW chronologies, respectively.

*Line 234: In the method the authors should explain how the individual proxies have been merged into a final multi-proxy methodology.*

The explanation has been added:

Reconstruction of MJJAS temperature was calculated with a multiple linear regression using as predictors $\delta^{13}C$, $\delta^{18}O$, MXD and TRW, according to formula:

$$MJJAS = 2.8708 * \delta^{13}C + 1.7606 * \delta^{18}O + 32.0744 * MXD + 5.5888 * TRW + 80.1884$$

*In Figure 2, it looks like the multi-centennial variability of the carbon isotope differs from the other proxies, driving the warm phase from 1650 to 1750 CE in the temperature reconstruction. This should be better explained, since this warm phase overlaps with the Little Ice Age, known to be a cold period in Europe and the Alps.*

The explanation has been added:

In the temperature reconstruction presented here, a time span of temperature increase that coincides with the Medieval Warm Period is evident and covers the years 975–1275 CE. In contrast, the period of the Little Ice Age, defined approximately between 1450 and 1850 CE, is shorter in the Schwarzensee temperature reconstruction and comprises the years 1475–1625 CE. However, on the basis of the research carried out, it is difficult to determine the reason. This shift might be influenced by the stable carbon isotope chronology, as it shows an increasing trend between 1600 and 1800 CE, maybe related to the fact that the stable isotope carbon chronology, in addition to temperature, depends strongly on the solar radiation of the summer months. The cause of the upward tendency in reconstructed temperature between 1650 and 1800 CE may also be moisture conditions as the time interval between 17th and 19th centuries was a wet period and all the chronologies used for reconstruction correlate to a greater or lesser extent with such climate parameters as precipitation, VAP, scPDSI, and SPEI indexes (Table 1).

The drivers may be also the range of months employed in the temperature reconstruction (May–September), which differs from many others previously performed studies, often taking into account only the summer period. The lack of perfect matching between the commonly used dates of Little Ice Age and the reconstruction derived from Lake Schwarzensee may also be the result of local environmental conditions that did not coincide with global temperature changes. The Intergovernmental Panel on Climate Change Third Assessment Report concluded that the Little Ice Age is largely independent regional climate change, rather than a globally synchronous increased glaciation and the timing and the areas affected by temperature decrease vary significantly (IPCC, 2022). The differences may also be explained by the combination of multiple proxies in the Schwarzensee reconstruction, in particular the usage of stable isotopes of carbon and oxygen, whose chronologies have the potential to capture more low-frequency variability compared to chronologies based on tree-ring width and maximum latewood density, which in turn, focus on the variability in the short-term domain. Therefore, it should be emphasised that the undeniable advantage and innovative aspect of the presented reconstruction is the application of four different chronologies owing to which obtained reconstruction has the unique potential to reflect long-term temperature trends.

*Line 330: Explain why the climate correlation time window analysis was narrowed for the final chronology but not for the individual proxies.*

In the revised version of the paper, a narrowed climate correlation time window analysis for individual proxies has also been added (Table 1). In particular, the narrowing of this window for the final chronology is due to the lack of correlation between the carbon chronology and the MIJJAS climate data in the 1960-2000 CE time interval and results from the large uncertainty in the meteorological measurements performed in the early 1800s.

*Line 424: Specify which panel of Figure 3 is being referred to.*

The explanation has been added:

In the case of the reconstruction carried out for the whole Alpine region (graph at the bottom of the Figure 6), the lowest temperatures were recorded in the 14th century, at the end of the 16th century and during the 17th century (Trachsel et al., 2012).

*Line 430-435. Nice result, but it should be noted that the high correlation pattern is probably driven by the high frequency, as all the other correlations are based on TRW and MXD, which point to a common high frequency signal. The difference may be in the low frequency or long-term trend. The new reconstruction seems to have a long temperature rise that the other reconstructions are not able to capture as they are based on TRW and MXD.*

The information has been added:

The high correlation rate obtained for this comparison points to a common short-term variability. The reason for this is most probably the employment of TRW and MXD measurements in both Schwarzensee and Alpine reconstructions. Nevertheless, owing to the addition of stable isotopes of carbon and oxygen the new reconstruction seems to reflect a low-frequency signal that the other studies are not able to detect as they are based on TRW and MXD data only.

**Figures:**

*A new introductory figure should be added showing the location of the study area and meteorological stations to provide an important geographical context.*

Map showing the geographical location of Schwarzensee Lake has been added (see above). However, the location of the meteorological stations has been omitted as their placement would have obscured the image of the map:

[Figure]

Location of the meteorological stations.

*In Figure 1, the y-axis scale should be included to allow proper interpretation of the data.*

The y-axis scale has been added:

[Figure]

**Figure 2: Schwarzensee chronologies: stable carbon isotope (δ¹³C–grey) chronology, stable oxygen isotope (δ¹⁸O–yellow) chronology, maximum latewood density (MXD–red) chronology, and ring-width (TRW–brown) chronology.**

*Figure 2 shows a very good agreement between the measurements and reconstruction, clearly demonstrating the proxy record accuracy.*

Thank you very much for this comment.

*Table 3 is a bit confusing in its current form. Consider adding the time windows analyzed in the first row to clarify the periods represented. Also use either R2 or r consistently for the correlation metrics.*

Required information has been added and table has been corrected:

**Table 4: Statistics calculated in Reconstats program used to evaluate the quality of the model applied for climate reconstruction – reduction of error (RE), coefficient of efficiency (CE), coefficient of determination ($R^2$), and Pearson correlation coefficient (r). Third column presents significances of the statistics in the 95 % and 99 % confidence intervals.**

| | | |
|---|---|---|
| Validation period (1840–1959): | RE=0.410 (p<0.001) | RE_95=0.046, RE_99=0.084 |
| Validation period (1840–1959): | CE=0.409 (p<0.001) | CE_95=0.023, CE_99=0.056 |
| Calibration period (1840–1899): | $R^2$=0.540 (p<0.001) | $R^2$_95=0.161, $R^2$_99=0.207 |
| Verification period (1900–1959): | $R^2$=0.546 (p<0.001) | $R^2$_95=0.065, $R^2$_99=0.111 |

*For Figure 4, adding y-axis scales and directly labelling the different temperature reconstructions on the plots would improve clarity and interpretation. Relating the observed temperature fluctuations to known climate periods, such as the Medieval Climate Anomaly and Little Ice Age, would provide helpful context for the trends. The reconstructions in Büntgen et al. 2011 and Büntgen et al. 2016 show the patterns as they are the same reconstruction, (Figure 2 Büntgen et al. 2016)*

*Figure 4, adding y-axis scales and directly labelling the different tree-ring based temperature reconstructions on the plots would improve clarity and interpretation. Relating the observed temperature fluctuations in these reconstructions to known climate periods, such as the Medieval Climate Anomaly and Little Ice Age, would provide helpful context. The reconstructions Büntgen et al. 2011 and Büntgen et al. 2016 show similar patterns, as they reflect the same tree-ring reconstruction (Figure 2 of the Büntgen et al. 2016 paper).*

Required corrections have been introduced:

[Figure]

**Figure 6: Comparison of the temperature reconstruction from Schwarzensee area with the other research. Red line– temperature anomalies for May–September (1901–2000 CE) for Schwarzensee Lake reconstructed on the basis of tree-ring width, tree-ring maximum latewood density and stable carbon and oxygen isotopes; yellow line–temperature anomalies for June–August (1961–1990 CE) for Austrian Alps reconstructed on the basis of tree-ring width (Büntgen et al., 2011); brown line–temperature anomalies for June–September (1901–2000 CE) for Swiss Alps reconstructed on the basis of tree-ring maximum latewood density (Büntgen et al., 2006); grey line–temperature anomalies for June–August (1901–2000 CE) for whole Alps reconstructed on the basis of tree-ring width, tree-ring maximum latewood density and lake sediments (Trachsel et al., 2012); black lines on each graph present the reconstructed values smoothed using spline fit with 50% variance cutoff at a wavelength of 25 years.**

---

## Author Comment (AC2)

Reply to the Reviewer 2:

Thank you very much for revising the manuscript, correcting it in detail, introducing the necessary changes and for all your comments. The review significantly improved the quality of article. We have taken into account all the suggested corrections and comments.

Revisions have been made in response to particular comments:

**General comments:**

*Hypotheses, background, and motivation for the conduct of this study are unclear. Why this study site was selected and why study was conducted? What is the advantage of using stable isotope proxies compared to classical tree-ring isotope analysis? Please provide citations relevant to the stable isotope studies.*

*The main output of the study highlighted cooling and warming periods with increasing air temperature trend since the beginning of the 20th century compared to the past.*

*The novelty (approach, region, and testing hypothesis) of this study compared to existing published studies must be specified. Citations and comparisons with existing studies in the region and globally should be provided (e.g., Kress et al. 2014; Büntgen et al., 2021; Churakova-Sidorova et al., 2022; Kuhl et al., 2024). It is unclear how the multi-proxy chronology was built and how it corresponds with other chronologies from Europe or with other proxies from the region like lake sediments.*

*Büntgen, U., Urban, O., Krusic, P.J. et al. Recent European drought extremes beyond Common Era background variability. at. Geosci. **14**, 190–196 (2021). https://doi.org/10.1038/s41561-021-00698-0*

*Churakova-Sidorova, O.V., Myglan, V.S., Fonti, M.V. et al. Modern aridity in the Altai-Sayan mountain range derived from multiple millennial proxies. Sci Rep **12**, 7752 (2022). https://doi.org/10.1038/s41598-022-11299-1*

*Kress, A., S. Hangartner, H. Bugmann, U. Büntgen, D. C. Frank, M. Leuenberger, R. T. W. Siegwolf, and M. Saurer (2014), Swiss tree rings reveal warm and wet summers during medieval times, Geophys. Res. Lett., 41, 1732–1737, doi:10.1002/ 2013GL059081.*

*Kuhl, E., Esper, J., Schneider, L. et al Revising Alpine summer temperatures since 881 CE. Clim Dyn (2024). https://doi.org/10.1007/s00382-024-07195-1*

Required information and explanations have been added:

Introduction chapter:

This article presents a long time scale temperature reconstructions for the Dachstein Mountains area localised in Northern Limestone Alps in Austria. The Alpine region has been used for palaeoclimatic studies since many decades. This is the result of beneficial factors that favour the undertaking of such research. The most important of these is the influence of the thermal conditions of the environment, which significantly limit the growth of plants and thus enable temperature changes to be reflected in the annual tree-rings. From the Alpine area predominantly originate dendroclimatic reconstructions based on tree-ring width and the maximum latewood

density. Among them are also multi-century analyses, extending back many hundreds of years (Schweingruber et al., 1987, 1988; Büntgen et al., 2005, 2006; Esper et al., 2007; Corona et al., 2010, 2011; Kuhl et al., 2024). On the other hand, chronologies based on stable isotope measurements have been constructed less frequently to date (Treydte et al., 2001; Gagen et al. 2006; Hafner et al., 2014). Moreover, they rarely reache back a thousand years (Kress et al., 2014). However, more than a millennium-old stable isotope studies are known from other territories (Churakova-Sidorova et al., 2022; Büntgen et al., 2021). In turn, in Alpine area reconstructions that use dendrochronological data together with other proxies, such as lake sediments, also occur (Trachsel et al., 2012). Sometimes historical informations are used to trace climate fluctuations in the past as well (Casty et al., 2005; Meier et al., 2007).

The reconstruction presented here was carried out on the basis of subfossil wood excavated from the small lake Schwarzensee and core samples bored from living trees growing around this lake. The region of Schwarzensee Lake provides an excellent location for dendroclimatic studies because it is a mountainous territory, under nature protection and therefore with negligible anthropogenic influence on tree growth. The choice of the study site is an novel aspect of the research, as palaeoclimatic reconstructions have not been carried out here before, and Schwarzensee Lake is in many ways a great place for this type of research.

Schwarzensee Lake is situated in mountain area and located almost at an elevation of regional timberline (47°31′ N, 13°49′ E, 1450 m.a.s.l.). Therefore, temperature is a dominant control influencing the length of vegetation period and the main factor limiting tree growth in this region. This strong interdependence creates an opportunity for the reconstruction of thermal conditions of the environment. A favourable aspect for palaeoclimatic studies is also the origin of wood from an area characterised by small territorial range. Because all the trees grew on the mountain slopes surrounding the Lake Schwarzensee, they were subjected during the course of their lives to the influences of the same weather conditions (Grabner et al., 2007). This ensures homogeneity of the climate signal recorded in annual tree-rings. Moreover, due to the fact that spruce wood dominated within the material sampled from the Schwarzensee Lake, it was possible to perform analyses on the basis on a single tree species only. This allowed to eliminate potential errors that can result from the various reactions of individual tree taxa to the impact of environmental factors. The extremely long meteorological record existing in the area is also a very beneficial situation. This is because temperature data dating back to the late 18th century allows for a more precise calibration when conducting palaeoclimatic research.

In order to make the best usage of wood for dendroclimatic research, various measurements have been performed during previous stages of the work. On this basis, four separate chronologies were built. Particular chronologies were constructed for the tree-ring width (TRW), the maximum latewood density (MXD) and for the content of stable carbon and oxygen isotopes within α-cellulose of individual tree-rings. The shortest of these, the oxygen stable isotope chronology dates back to the eight hundredth year CE. This therefore enabled temperature reconstructions to be carried out over a common time interval of up to 1,200 years. This is also an extremely favourable circumstance, as it makes it possible to place contemporary climatic changes in the context of centuries-old fluctuations existing in the past.

The objective of the presented study was to reconstruct the thermal conditions of the environment in the area extending around Lake Schwarzensee, in a period covering the common time range of previously created chronologies. Palaeoclimatic research has been undertaken because it is still necessary to provide more and more precise multi-century climatic data (Kuhl et al., 2024). An innovative aspect of this work is that it presents the first temperature reconstruction of over a thousand years from the Alpine area, based on several proxies: tree-ring width, maximum latewood density, and stable carbon and oxygen isotopes. Reconstruction demonstrated here differs from many previous studies of this type, as most of them consider a much narrower range of months and concern only the temperature of summer season. In contrast, reported results deliver a reconstruction of the temperature of the entire vegetation period (the months from May to September). An undeniable advantage of the analyses presented here is the employment of a number of various proxies, including carbon and oxygen stable isotope chronologies, which allow low-term weather trends to be preserved. In comparison with them, the most commonly used tree-ring width or maximum latewood density reflect more temperature fluctuations in the high-frequency domain. Therefore, the applied approach makes it possible to trace thermal changes more accurately over millennial time span and, consequently, to better place contemporary climate warming in historical framework. Results demonstrated here has therefore the potential to improve temperature reconstruction and its temporal reliability. This constitutes an exceptionally favourable situation that distinguishes this work from most previous studies conducted using chronologies of tree-ring width and maximum density of latewood.

In order to most accurately determine the weather in the past, during conducted studies a multi-proxy approach was utilised. A combination of several proxies was aimed to increase the amount and the quality of palaeoenvironmental information derived from tree-ring data. Due to the employment of several tree-ring parameters to reconstruct a single environmental variable the reduction of the noise associated with each of the proxies was also obtained and the enhancement of the climatic signal was gained as well (McCarroll et al., 2003).

Materials and methods chapter:

Reconstruction of MJJAS temperature was calculated with a multiple linear regression using as predictors $\delta^{13}C$, $\delta^{18}O$, MXD and TRW, according to formula:

MJJAS=$2.8708*\delta^{13}C$ +$1.7606*\delta^{18}O$ +32.0744*MXD +5.5888*TRW +80.1884

Results and discussion chapter:

In the obtained reconstruction the effect of volcanic events which could significantly reduce temperature value is also evident. Despite the fact that the area of Schwarzensee Lake is located far from volcanically active regions, however, severe eruptions periodically alter weather conditions, even on a global scale. Taking into account only strong events characterized by an explosivity index equal to or greater than 4 (Siebert and Simkin, 2002) and the fifty coldest years in the entire Schwarzensee temperature reconstruction, 13 of these are could be related to the impact of volcanoes. Although it is rather difficult to unambiguously ascribe cold years to a particular eruption however, probably temperature decrease which appeared in years 1020, 1032, 1040, 1281, 1290, 1587, 1813, 1814, 1815, 1816, 1912, 1913, 1933 were triggered by them. Observed interrelationship is apparent especially for the younger part of the data because for earlier time periods there is a lot of uncertainty due to imprecise dating of the oldest volcanic eruptions. The above comparison is based on the data available on the website https://volcano.si.edu/.

Other global climate fluctuation, such as the Medieval Warm Period or the Little Ice Age, can also be identified in the reconstruction from Lake Schwarzensee. The time span of temperature increase that coincides with the Medieval Warm Period is evident and covers the years 975–1275 CE. In contrast, the period of the Little Ice Age, defined approximately between 1450 and 1850 CE, is shorter in the Schwarzensee temperature reconstruction and comprises the years 1475–1625 CE. However, on the basis of the research carried out, it is difficult to determine the reason. This shift might be induced by the stable carbon isotope chronology, as it shows an increasing trend between 1600 and 1800 CE, maybe related to the fact that the stable isotope carbon chronology, in addition to temperature, depends strongly on the solar radiation of the summer months. The cause of the upward tendency in reconstructed temperature between 1650 and 1800 CE may also be moisture conditions as the time interval between 17th and 19th centuries was a wet period and all the chronologies used for reconstruction correlate to a greater or lesser extent with such climate parameters as precipitation, VAP, scPDSI, and SPEI indexes (Table 1).

The drivers may be also the range of months employed in the temperature reconstruction (May–September), which differs from many others previously performed studies, often taking into account only the summer period. The lack of perfect matching between the commonly used dates of Little Ice Age and the reconstruction derived from Lake Schwarzensee may also be the result of local environmental conditions that did not coincide with global temperature changes. The Intergovernmental Panel on Climate Change Third Assessment Report concluded that the Little Ice Age is largely independent regional climate change, rather than a globally synchronous increased glaciation and the timing and the areas affected by temperature decrease vary significantly (IPCC, 2022). The differences may also be explained by the combination of multiple proxies in the Schwarzensee reconstruction, in particular the usage of stable isotopes of carbon and oxygen, whose chronologies have the potential to capture more low-frequency variability compared to chronologies based on tree-ring width and maximum latewood density, which in turn, focus on the variability in the short-term domain. It should be emphasised that the undeniable advantage and innovative aspect of the presented research is the application of four different chronologies owing to which Schwarzensee reconstruction has the unique potential to reflect long-term temperature trends.

Nevertheless, obtained results reflects the temperature conditions at a relatively small territory localised close to Lake Schwarzensee. This is due to the fact that all of the material used to the construction of individual chronologies originated from a very small area surrounding the lake. This is a beneficial factor for conducting palaeoenvironmental research because it ensures homogeneity of the climate signal. At the same time, however, it also causes that reconstruction has a local scale. This is clearly visible especially in relation to the Alps, because in this region there is a large spatial variability of weather conditions (Böhm et al., 2001). Therefore, the temperature value and the sum of precipitation can vary strongly within the sites situated close to each other. In order to identify whether obtained reconstruction represents climate trends that are coherent over a larger spatial scale, the comparison between obtained results and other tree-ring based temperature reconstructions from nearby Alpine regions was carried out (Fig. 6).

Conclusion chapter:

A lot of various factors favoured the possibility of undertaking a climatic reconstruction on the basis of Schwarzensee wood. These were the employment of a single plant species (this made it possible to avoid potential errors that can result from the various reactions of individual tree taxa to the impact of environment), the origin of the wood from a spatially small region (which ensured homogeneity of the climate signal), the mountainous character of the area (with negligible anthropogenic influence and with temperature as a dominant control limiting tree growth), the extremely long sequence of climatic data available for this site (what allowed for a more precise calibration during palaeoclimatic research). The long time span of the reconstruction, which covers 1,200 years, was also a very important aspect.

The main output of the study highlight cooling and warming periods during reconstruction time-span as well as increasing air temperature trend since the beginning of the 20th century compared to the past.

**Specific comments:**

**Title:** *The title can be shortened. Suggestion: "A 1200-year multi-proxy May-September temperature reconstruction from the Austrian Alps" or even revised based on the findings and case study.*

The title has been changed:

An 1200-year multi-proxy May-September temperature reconstruction derived from tree-rings – a case study from the Schwarzensee (Austrian Alps)

**Abstract:**

*L. 15-16 it will be good to provide clarification at the beginning of the manuscript about averaged May-September air temperature reconstruction.*

Required information has been added:

Averaged May-September air temperature reconstruction was carried out on the basis of a centuries-long dendrochronological scales from Austrian Alps.

*L. 16 and throughout the whole manuscript "growth-ring width" replace with tree-ring width; "maximum density of latewood" replace with maximum latewood density; "stable*

*isotope content of carbon and oxygen" replace with stable carbon and oxygen isotopes or isotopic composition of stable carbon and oxygen isotopes. Please also add a small delta symbol for stable carbon and oxygen isotopes. Basic dendrochronological characteristics should be provided in a standard way: as wood density, tree-ring (see Methods of Dendrochronology https://link.springer.com/book/10.1007/978-94-015-7879-0, E. R. Cook &L.A. Kairiukstis, 1990).*

Required corrections have been made.

*It is unclear from the abstract how the reconstruction of May-September was performed. Is it 52% of temperature variance explained in combined multi-proxy reconstruction based on tree-ring width, stable carbon and oxygen isotopes, and latewood density? How much (in %) is it explained by each parameter (carbon, oxygen, tree-ring width, maximum latewood density)? Please specify.*

Required explanation have been added:

Reconstruction of May–September temperature was calculated with a multiple linear regression using as predictors $\delta^{13}C$, $\delta^{18}O$, maximum latewood density and tree-ring width. Coefficient of determination received during calculation of regression model shows that the proportion of temperature variance explained by combined multi-proxy reconstruction based on tree-ring width, stable carbon and oxygen isotopes, and latewood density accounted for 52% in the period common for the tree-ring chronologies and meteorological data ($R^2$=0.518, p<0.001). In particular, temperature variance is explained in 0.161% by carbon chronology, in 0.274% by oxygen chronology, in 0.397% by maximum latewood density chronology and in 0.074 % by tree-ring width chronology.

*L. 22-23 "..the statistic is good.." Please provide numbers and confirm.*

Required information has been added:

The 'effective correlation' between the estimated and measured values of May–September temperature accounted for 0.72. To evaluate the similarity between the observed and estimated data reduction of error (RE) and coefficient of efficiency (CE) were also calculated. Obtained RE was equal to 0.410 (p<0.001), and CE amounted to 0.409 (p<0.001). The positive and relatively high value of RE and CE coefficents demonstrates strong reconstruction skills of the chronologies and is an evidence for a valid regression model.

*L.25 "The strongest increasing trend» How strong and significant this trend is compared to the past? Please specify and provide values or estimates in %.*

Required information has been added:

The mean reconstructed temperature from 1900–2000 CE is 24.84° C, and is 0.5° C higher than the mean from 800–1899 CE, and the six warmest years in the entire reconstruction are from the 20th century.

**1. Introduction**

*More citations and comparisons with other studies from the region and the globe are recommended.*

Required information has been added:

This article presents a long time scale temperature reconstructions for the Dachstein Mountains area localised in Northern Limestone Alps in Austria. The Alpine region has been used for palaeoclimatic studies since many decades. This is the result of beneficial factors that favour the undertaking of such research. The most important of these is the influence of the thermal conditions of the environment, which significantly limit the growth of plants and thus enable temperature changes to be reflected in the annual tree-rings. From the Alpine area predominantly originate dendroclimatic reconstructions based on tree-ring width and the maximum latewood density. Among them are also multi-century analyses, extending back many hundreds of years (Schweingruber et al., 1987, 1988; Büntgen et al., 2005, 2006; Esper et al., 2007; Corona et al., 2010, 2011; Kuhl et al., 2024). On the other hand, chronologies based on stable isotope measurements have been constructed less frequently to date (Treydte et al., 2001; Gagen et al. 2006; Hafner et al., 2014). Moreover, they rarely reach back a thousand years (Kress et al., 2014). However, more than a millennium-old stable isotope studies are known from other territories (Churakova-Sidorova et al., 2022; Büntgen et al., 2021). In turn, in Alpine area reconstructions that use dendrochronological data together with other proxies, such as lake sediments, also occur (Trachsel et al., 2012). Sometimes historical informations are used to trace climate fluctuations in the past as well (Casty et al., 2005; Meier et al., 2007).

*L 35-40 Please provide citations*

Required citations has been added:

Wood is an excellent archive of environmental variability because plant tissues record climate conditions existing during their growth. Weather factors affect the intensity of physiological processes and in this way influence the tree-ring width, wood density and isotopic signal in wood (Hughes et al., 2010). The changes of physico-chemical properties of tree-rings are influenced by solar radiation, temperature, precipitation, soil moisture, and atmospheric humidity. Fortunately, the dependencies between environmental forces and internal mechanisms taking place in plants are well understood (Fritts, 1976; Schweingruber et al., 1996; Siegwolf et al., 2022). As a result, measurements of various wood parameters within individual tree-rings allow tracking climate variables with an annual resolution and in long-term time intervals (Speer, 2010).

*The "Introduction" and "Materials and Methods" sections are mixed up with the introducing region in the introduction part.*

A brief description and introduction of the region in the Introduction chapter was necessary because it enabled to explain the motivation for the study, the purpose of the research and the choice of the study area.

*L. 59 …ring width (TRW) replaced with tree-ring width (TRW)*

Required correction has been made:

Particular chronologies were constructed for the tree-ring width (TRW), the maximum latewood density (MXD) and for the content of stable carbon and oxygen isotopes within individual tree-rings.

**2. Material and Methods**

*L. 100 ".. were omitted during densitometric and isotopic research" replace with .." were excluded from densitometric and stable isotope measurements".*

Required correction has been made:

Therefore, the oldest samples were more destroyed and thus they were excluded from densitometric and stable isotope measurements.

*L. 110-115 Please provide an original citation to X-ray densitometrical tool by Schweingruber FH and Briffa KR 1995 and Schweingruber FH 1988. The citation to Kłusek and Grabner, 2016 is not the original one.*

Required correction has been made:

Densitometric measurements were carried out according to the methodology described by the Schweingruber et al. (1978), Schweingruber (1988), Schweingruber and Briffa (1995).

*L. 140-141 "Then, the material obtained from four trees was pooled (combined) for a particular year." This sentence contradicts the sentence above that 5 living trees were used for the stable isotope analyses (L. 139). Please clarify how many trees as a pooled material were used for the stable isotope analyses.*

Required explanation has been added:

For stable isotope analyses 51 subfossil stem discs and samples from five living trees were prepared, selected so that in each successive year the material came from a total of four trees (not all cores covered the period 1800-2000 CE).

[Figure]

The time spans of samples. Each of the four horizontal bars is divided into segments corresponding to individual trees. Colours indicate particular trees—a tree is darker or lighter than the previous one.

*L. 140-150 The method for cellulose extraction described is originally developed by Loader 1997 https://doi.org/10.1016/S0009-2541(96)00133-7. If this protocol and description were modified, this should be explicitly mentioned.*

Required correction has been added:

The procedure of α-cellulose isolation was conducted in accordance with the modified Green (1963) method, following the protocol described by Loader et al. (1997) and adapted for Schwarzensee wood (Kłusek and Pawełczyk, 2014).

*L. 143 if cellulose was extracted and analyzed, it should be mentioned at the beginning of the article.*

Required information has been added:

Abstract:

Chronologies of tree-ring width, maximum latewood density and stable carbon and oxygen isotopes (based on α-cellulose) were applied for the research.

Introduction chapter:

Particular chronologies were constructed for the tree-ring width (TRW), the maximum latewood density (MXD) and for the content of stable carbon and oxygen isotopes within α-cellulose of individual tree-rings.

*L. 164 Why was analysis performed until 2000, while samples were available until 2015? The 15 years can make a big difference. Please clarify.*

Required explanation has been added:

Moreover, 80 spruce and larch samples were bored from the trees growing around the lake in 1999 (Grabner et al., 2007). However, along with the development of subsequent chronologies there was a need for the new samples. Therefore, additional 40 cores from the living trees was collected in years 2009 and 2015. Nevertheless in this case, the increment cores taken had an smaller inner diameter than during first sampling carried out using dry wood borers. Hence, the climatic reconstruction presented here reaches to the year 2000 CE, as some of the measurements were conducted using cores of bigger diameter only due to the larger amount of wood available for preparation.

*L. 165-166 Please clarify what kind of correction was performed for stable carbon isotopes (pin? δ13C atm CO2 only?) and if any corrections were applied for oxygen isotope chronology.*

Required explanation has been added:

For the period of 1850–2000 CE, an additional correction was introduced because of the anthropogenic increase of the atmospheric $CO_2$ concentration and related lowering of $\delta^{13}C$ of air. Therefore, the raw $\delta^{13}C$ measurements were adjusted to a pre-industrial atmospheric $\delta^{13}C$ base value of −6.4‰ using simple addition. It was done on the basis of the published annual records of the stable carbon isotope ratios of atmospheric carbon dioxide (McCarroll et al., 2009). Moreover, in order to compensate the change in the ecophysiological response to the rapidly rising concentrations of $CO_2$, the second correction, known as the PreINdustrial or PIN correction, was applied to the fossil fuel corrected data in the period of 1820–2000 CE (McCarroll et al., 2009; Konter et al., 2014). In turn, for the stable oxygen isotopes chronology no correction was used.

*L. 167-170 Why climate analysis was performed with temperature only and not with or additionally with other climatic parameters (e.g., precipitation, vapor pressure deficit, relative humidity)? Please clarify.*

In addition to correlations with temperature, the climate analysis has been extended to examine relationships with other climatic parameters and presented in Table 1:

| | δ¹³C | | | | | | δ¹⁸O | | | | | | MXD | | | | | | TRW | | | | | |
|---|---|---|---|---|---|---|---|---|---|---|---|---|---|---|---|---|---|---|---|---|---|---|---|---|
| | TEM | PRE | VAP | PDSI | SPEI | RAD | TEM | PRE | VAP | PDSI | SPEI | RAD | TEM | PRE | VAP | PDSI | SPEI | RAD | TEM | PRE | VAP | PDSI | SPEI | RAD |
| January | -0,22 | 0,01 | -0,18 | 0,11 | -0,05 | 0,18 | 0.05 | 0.10 | 0.08 | 0.15 | 0.07 | 0.00 | 0.12 | -0.10 | 0.09 | 0.07 | -0.10 | 0.02 | -0.06 | 0.18 | -0.07 | 0.14 | 0.17 | -0.14 |
| February | 0,01 | **0,22** | 0,04 | 0,20 | 0,20 | -0,11 | -0.01 | 0.08 | 0.03 | 0.10 | 0.00 | 0.06 | -0.02 | 0.07 | -0.02 | 0.03 | -0.01 | 0.04 | 0.12 | 0.09 | 0.13 | 0.19 | 0.11 | -0.08 |
| March | -0,12 | 0,09 | -0,07 | 0,11 | 0,13 | -0,05 | 0.02 | -0.09 | 0.02 | 0.01 | 0.01 | -0.03 | 0.13 | 0.09 | 0.04 | -0.02 | 0.07 | 0.04 | -0.09 | 0.07 | 0.01 | 0.11 | 0.06 | -0.16 |
| April | 0,03 | 0,10 | 0,08 | 0,18 | 0,03 | 0,19 | 0.17 | -0.04 | 0.15 | -0.04 | -0.06 | 0.09 | **0.26** | -0.11 | **0.22** | -0.08 | -0.17 | **0.27** | 0.00 | 0.08 | 0.07 | 0.15 | 0.07 | -0.04 |
| May | 0,14 | -0,20 | -0,04 | 0,09 | -0,25 | 0,17 | **0.30** | -0.11 | **0.24** | -0.09 | **-0.21** | **0.23** | **0.35** | **-0.21** | **0.22** | -0.14 | **-0.31** | 0.17 | **0.16** | -0.14 | 0.10 | 0.06 | -0.12 | 0.17 |
| June | 0,19 | -0,12 | 0,04 | 0,00 | -0,15 | **0,28** | 0.10 | -0.04 | -0.07 | -0.16 | -0.06 | 0.11 | 0.13 | 0.09 | 0.17 | -0.09 | 0.10 | -0.15 | **0.29** | -0.10 | 0.17 | -0.05 | -0.16 | **0.35** |
| July | **0,42** | -0,15 | **0,35** | -0,08 | **-0,26** | **0,54** | **0.50** | **-0.52** | 0.19 | **-0.35** | **-0.54** | **0.65** | **0.45** | **-0.20** | **0.30** | -0.16 | **-0.32** | **0.39** | **0.23** | -0.08 | 0.12 | -0.05 | -0.11 | **0.27** |
| August | **0,51** | **-0,30** | 0,25 | -0,18 | **-0,27** | **0,41** | **0.35** | -0.19 | 0.10 | **-0.37** | **-0.19** | **0.25** | **0.67** | -0.13 | **0.51** | **-0.22** | **-0.24** | **0.27** | 0.02 | -0.12 | -0.07 | -0.11 | -0.08 | 0.06 |
| September | -0,05 | 0,01 | -0,08 | -0,10 | -0,11 | -0,03 | 0.06 | -0.10 | 0.09 | **-0.33** | -0.07 | 0.05 | **0.48** | **-0.37** | **0.36** | **-0.30** | **-0.38** | **0.31** | 0.09 | -0.09 | 0.05 | -0.11 | -0.13 | 0.08 |
| October | 0,03 | 0,09 | 0,00 | -0,03 | 0,07 | -0,01 | 0.01 | -0.06 | -0.03 | **-0.31** | -0.04 | 0.15 | -0.02 | 0.08 | -0.08 | -0.19 | 0.03 | **0.19** | 0.11 | -0.07 | -0.01 | -0.16 | -0.11 | 0.10 |
| November | -0,23 | 0,20 | -0,09 | 0,11 | 0,22 | -0,17 | -0.11 | 0.13 | -0.16 | -0.18 | 0.08 | -0.05 | 0.19 | 0.10 | 0.09 | -0.11 | 0.13 | **0.26** | 0.04 | 0.07 | 0.00 | -0.13 | 0.08 | 0.06 |
| December | 0,11 | -0,08 | 0,16 | 0,07 | -0,04 | -0,07 | -0.03 | 0.10 | 0.07 | -0.10 | 0.08 | -0.13 | 0.10 | 0.11 | 0.03 | 0.02 | 0.13 | -0.02 | -0.03 | -0.18 | -0.04 | -0.14 | -0.12 | -0.03 |

**Table 1. Correlations with climate parameters. Bootstrap correlation were calculated in DENDROCLIM 2002 between climate data: mean monthly temperature (TEM), total monthly precipitation (PRE), mean monthly sunshine duration (RAD), monthly averages of vapour pressure (VAP), self-calibrating Palmer Drought Severity Index (PDSI), and Standardized Precipitation-Evapotranspiration Index (SPEI), and tree-ring parameters: Schwarzensee tree-ring width (TRE), maximum latewood density (MXD), stable carbon (δ¹³C), and oxygen (δ¹⁸O) isotope chronologies. Significant values at the 0.05 levels are marked in bold. Analyses were undertaken over a common period 1901–2000 CE.**

Obtained chronologies of the tree-ring width, maximum latewood density, stable carbon and oxygen isotopes were compared with climatic records. For this purpose, various meteorological parameters were applied. The mean monthly temperature (TEM) data originated from neighbouring meteorological stations and was gridded for the area of Schwarzensee Lake (Auer et al., 2005, 2007; Efthymiadis et al., 2006; Chimani et al., 2013). Homogenised mean monthly sunshine duration (RAD) derived from the meteorological station of Kremsmünster (Auer et al., 2007). Total monthly precipitation (PRE) dataset was based on homogenized precipitation series from adjacent meteorological stations and was gridded at 10-minute resolution (Efthymiadis et al., 2006). Self-calibrating Palmer Drought Severity Index (scPDSI) was calculated using as input the interpolated monthly precipitation and temperature observations with a resolution of 10 minutes longitude by 10 minute latitude (van der Schrier et al., 2007). Vapour pressure (VAP) record contained the spatial monthly averages calculated using area-weighted means for Austria. It was based on the CRU TS high-resolution multivariate gridded data set (Harris at al., 2020). The Standardized Precipitation-Evapotranspiration Index (SPEI) came from the Global SPEI database which offers long-time, robust information about drought conditions with a 0.5 degrees spatial resolution and a monthly time resolution. It is based on monthly precipitation and potential evapotranspiration data from the Climatic Research Unit of the University of East Anglia. This database provides SPEI time-scales between 1 and 48 months and SPEI time-scale used during presented study was 1 month (Beguería et al., 2014). In total, the meteorological records employed for this research encompassed monthly averaged data for temperature (1780–2000), solar radiation (1884–2000), monthly sums of precipitation (1800–2000), monthly averaged self-calibrating Palmer Drought Severity Index (1800–2000), vapour pressure (1901–2000), and Standardized Precipitation-Evapotranspiration Index (1901–2000).

*L. 171-172 What is the reason for taking the overlap period from September of the previous year to October of the current one? In this case correlation coefficients due to overlap will be higher, but not necessarily will provide reasonable eco-physiological information, which is supposed to be derived from the stable isotopes and tree-ring parameters. Were the 14 months to all tree-ring width and stable isotope parameters applied? In Table 1 later is even from January of the previous year to December of the current one? Please explain.*

Required correction has been made:

Climate-growth relationships were determined by examining the correlation coefficient (r) between measured tree-ring proxies and climatic record. Coefficients were calculated during a 12 month period beginning in January and ending in December of the current year, in the time interval 1901–2000 CE for all of tree-ring and stable isotope parameters. For the temperature additionally the time span 1780–2000 CE was presented.

The range of months taken to calculate the correlation with climate parameters has been corrected. The presentation of a wider range of months in the previous version of the manuscript was informative only. The incorporation of a larger number of months did not affect the results obtained, as only the values of the correlation coefficient were presented. The values of the correlation coefficient are independent of the number of months taken into account in the calculations as the program calculates correlation coefficient separately for each month (in contrast to the response function, for which the number of months taken into account has a significant impact).

**3. Results and Discussion**

*L. 182 „ ...various physicochemical parameters of wood..” What is ment here? Stable isotopes as biogeochemical tools?*

Required correction has been made:

On the basis of the conducted research, the chronologies presenting various parameters of wood were constructed, namely chronology of maximum latewood density and the corresponding chronology of tree-ring width as well as stable carbon and oxygen isotope chronologies.

*L. 184 It is unclear why until 2000 only, while living trees were collected until 2015.*

Required explanation has been added:

The sampling took place in the summer of the year 1999. During the works, 211 trees were lifted from the bottom of the lake, 66 % of them were spruce (*Picea abies* (L.) Karst.), 21 % larch (*Larix decidua* Mill.) and 13 % belonged to stone pine (*Pinus cembra* L.). Stem discs were cut out from each log and afterwards, sampled trunks were submerged back in the lake's waters. Moreover, 80 spruce and larch samples were bored from the trees growing around the lake in 1999 (Grabner et al., 2007). However, along with the development of subsequent chronologies there was a need for the new samples. Therefore, additional 40 cores from the living trees was collected in years 2009 and 2015. Nevertheless in this case, the increment cores taken had an smaller inner diameter than during first sampling carried out using dry wood borers. Hence, the climatic reconstruction presented here reaches to the year 2000 CE, as some of the measurements were conducted using cores of bigger diameter only due to the larger amount of wood available for preparation.

*L. 187-190 should be moved to the Material and Methods section.*

Required correction has been made:

In order to determine the suitability of chronologies for dendroclimatic studies, the commonly used Expressed Population Signal (EPS) was calculated (Wigley et al., 1984; Briffa and Jones, 1990). Usually, for EPS a threshold value of 0.85 is accepted, above which a chronology is assumed to represent the population signal with sufficient confidence for reconstruction purposes (Wigley et al., 1984; Briffa and Jones, 1990). Signal strength of the TRW and MXD chronologies was assessed using running EPS computed over a moving window of 50 year with 49 year overlap, to reflect the variability of EPS back in time.

*L. 216 What about August?*

Correlation with August was not significant:

The best results for the tree-ring width chronology were noted for May, June and July. The finding of high correlations between TRW and temperature in June and July agrees with previous work such as Leal et al. 2007.

*Results of Table 1 showed that TRW negatively correlated with summer temperatures of the previous year. Please explain why the statistical analysis was performed from January of the previous year to December of the current one. Why not consider from September of the previous year to August of the current one? Overlaps with the months can give better statistical values but do not make any eco-physiological sense. Otherwise, please explain.*

The presentation of a wider range of months in the previous version of the manuscript was informative only. The range of months taken to calculate the correlation with climate parameters has been corrected:

Climate-growth relationships were determined by examining the correlation coefficient (r) between measured tree-ring proxies and climatic record. Coefficients were calculated during a 12 month period beginning in January and ending in December of the current year, in the time interval 1901–2000 CE for all of tree-ring and stable isotope parameters. For the temperature additionally the time span 1780–2000 CE was presented.

*Why only the temperature parameter was considered and not other climatic parameters like precipitation, vapor pressure deficit, or relative humidity? Did you check correlations with other climatic parameters? If yes, please provide information in supplementary material or at least check it or cite, if it was published before.*

Required correlation analysis has been added in Table 1. Moreover, description has been added:

The obtained chronologies of tree-ring width, maximum latewood density, carbon and oxygen stable isotopes were compared with climatic variables. For this purpose, correlation coefficients were calculated between the individual chronologies and temperature, precipitation, solar radiation, vapour pressure, self-calibrating Palmer Drought Severity Index and Standardized Precipitation-Evapotranspiration Index variables. In order to be able to better compare the results obtained, all calculations were carried out over the period 1901–2000 CE, as this was the maximum range of the chronologies used for the study and part of the climate data. The results are presented in Table 1. As an effect of this calculations, it was observed that weather parameters influenced individual chronologies in different way and with various strength. Because temperature fluctuations over time are the most interesting parameter in the context of ongoing climate change, a reconstruction of the thermal conditions of the environment was therefore carried out in the work presented here. Temperature is also the only parameter that significantly affects all analysed proxies, so all chronologies from the Schwarzensee Lake area could be used for its reconstruction. This decision made it also possible to apply a multi-proxy approach in the study. A combination of several proxies was aimed to increase the amount and the quality of palaeoenvironmental information derived from tree-ring data. Due to the employment of several tree-ring parameters to reconstruct a single environmental variable it was possible to reduce the noise associated with each of the proxies and to enhance the climate signal. Moreover, the incorporation of stable isotope chronologies into research allowed better identification of reconstructed long-term trends, compared to studies using only TRW or MXD, which better reflect the changes in the high frequency domain.

*L. 245 – References are needed.*

Required references have been added:

The width of tree-rings depends on various weather parameters. Environmental conditions affect all physiological processes taking place in the tree. Climate factors control net assimilation and photosynthesis rate, level of cellular respiration, plant nutrition, hormone functions, transpiration and stomatal conductance as well as all processes of growth and development on cellular level and at the scale of the whole plant. They significantly impact division, enlargement and maturation of wooden cells (Fritts, 1976).

*L. 214 "Péclet effect.." please provide relevant citations (e.g., Craig and Gordon 1965)*

Required references have been added:

Nevertheless, this enlarged concentration is counteracted by the Péclet effect which consists in the convection of isotopically lighter source water through the transpiration flow (Craig and Gordon, 1965; Barbour et al., 2004; Gagen et al., 2011).

*L. 310-314 Not all parameters captured temperature signal from May to September. This should be taken into consideration and discussed.*

Required explanation has been added:

To precisely examine this influence, in the course of preliminary studies conducted with the newly constructed Schwarzensee chronologies, various growing season intervals were tested when determining relationships with climatic data. As a result, it was decided to reconstruct temperature of May–September months, as both the carbon and oxygen stable isotope chronologies as well as the maximum latewood density chronology show significant correlations with temperature within the range of these months. The exception to this is June temperature, for which the MXD chronology has a low and insignificant coefficient value (Table 2). Better linkage appears, however, if a shorter period of weather data is considered (skipping first part of the 19th century). This is undoubtedly justified by the lower quality of meteorological observations dating before to this time. In addition, there is also a significant connection between the June temperature and the formation of light rings, which are characterised by a substantial reduction in the maximum density of latewood. In turn, light ring genesis is associated with an intense short-term events and these growth-ring disturbances could be used as bioindicator for extreme cooling (see Kłusek et al., 2015). This therefore proves that the brief and strong temperature decrease during June, with which the formation of light rings is associated, has a strong impact on the maximum density of latewood.

The second case for which the temperature dependence does not extend over the entire May-September period is the tree-ring width chronology, which has significant correlations in the months of May–July, but its correspondence with the temperature of August–September is not significant. This is most probably due to the relatively low values of the correlation coefficient obtained for the TRW chronology. However, this coefficient for the averaged MJJAS temperature already has a significant value of 0.354, indicating the usefulness of TRW chronology in reconstructing the entire assumed range of months. Nevertheless, as this correlation is relatively weak compared to the other chronologies, the contribution of tree-ring widths to the reconstruction is smallest, as shown in the regression equation presented above.

*L. 349 it is unclear why the period from 1960 to 2000 was not included in reconstruction. If there is a reason, please explain.*

Required explanation has been added:

In the second half of the twentieth century, the values of tree-ring proxies are disturbed by the influence of human activity (Fonti et al., 2010). Moreover, during this time human induced increase in temperature level is also evident (IPCC, 2022). Both of these reasons cause the changes in the relationship between the tree-ring parameters and climatic factors. This phenomenon was observed directly for Schwarzensee chronologies (Kłusek et al., 2019). As a result, the usage of data burdened with 'anthropogenic noise' would have disadvantageous impact on the reconstruction authenticity. Therefore, to avoid possible errors, for the validation with meteorological record the period starting since 1840 CE and ending in the year 1959 CE was selected. Further cause why the years from 1960 CE to 2000 CE were not included into reconstruction was the lack of correlation between the stable carbon isotope chronology and the MIJJAS climate data during this time interval.

*L. 388 "These results reflects the temperature conditions at a relatively small territory localised close to Lake Schwarzensee.". It would make sense to rephrase the title as a case study from the Austrian Alps or specify A 1200-year Schwarzensee May-September air temperature reconstruction derived from tree-ring parameters and stable isotopes.*

The title has been changed:

An 1200-year multi-proxy May-September temperature reconstruction derived from tree-rings – a case study from the Schwarzensee (Austrian Alps)

**Figures:**

*Figure 1: It will be interesting to see not a standard deviation, but rather raw data. Here is also unclear if is it z-scored data or not. If yes, should be specified. Please add the symbol small delta (δ) to 13C and 18O.*

Required corrections have been made:

[Figure]

**Figure 2: Schwarzensee chronologies: stable carbon isotope (δ¹³C–grey) chronology, stable oxygen isotope (δ¹⁸O–yellow) chronology, maximum latewood density (MXD–red) chronology, and ring-width (TRW–brown) chronology.**

Figure 2: It is unclear how multi-proxy reconstruction was obtained and / or for which chronology the comparison with averaged May-September air temperature is shown in Fig 2. Please clarify in the Figure legend.

Required corrections have been made:

**Figure 3: Temporal agreement between observed mean May–September temperature (meteorological data) and Schwarzensee reconstruction based on tree-ring width, maximum latewood density, and stable carbon and oxygen isotopes. Measured (red line) and predicted (black line) value for averaged MJJAS temperature during the time interval 1780–2000 CE is presented.**

Figure 3: capture: Reconstats program – please add a citation.

Required citation has been added:

**Figure 4: Schwarzensee May-September temperature reconstruction (black line) and the reconstructed values smoothed using spline fit with 50% variance cutoff at a wavelength of 25 years (red line). Grey shade represents lower and upper 95% confidence interval values calculated in Reconstats program (Macias-Fauria et al., 2012).**

**Table:**

Table 1: Please add symbol small delta (δ) to 13C and 18O.

Required corrections have been made:

| | $\delta^{13}C$ | $\delta^{18}O$ | MXD | TRW |
|---|---|---|---|---|
| January | **0.167** | 0.048 | 0.118 | 0.062 |
| February | **0.141** | 0.062 | 0.015 | 0.090 |
| March | 0.104 | 0.118 | **0.161** | 0.008 |
| April | 0.042 | 0.027 | **0.177** | -0.043 |
| May | **0.214** | **0.211** | **0.244** | **0.141** |
| June | **0.135** | **0.187** | 0.054 | **0.399** |
| July | **0.363** | **0.508** | **0.323** | **0.274** |
| August | **0.420** | **0.302** | **0.505** | 0.051 |
| September | **0.158** | **0.151** | **0.359** | 0.135 |
| October | **0.144** | 0.047 | 0.070 | 0.047 |
| November | 0.045 | 0.010 | **0.184** | **0.142** |
| December | 0.066 | -0.018 | **0.134** | 0.012 |
| MJJAS | **0.459** | **0.483** | **0.537** | **0.354** |

**Table 2: Bootstrap correlation coefficients calculated in DENDROCLIM 2002 between mean monthly temperature and stable carbon isotope chronology ($\delta^{13}C$), stable oxygen isotope chronology ($\delta^{18}O$), maximum latewood density chronology (MXD), and ring-width chronology (TRW). Significant values at the 0.05 level are marked in bold. Calculations were performed over the entire period of available meteorological data – 1780–2000 CE. MJJAS – temperature averaged for May–September period.**